# TIME CAN INVALIDATE ALGORITHMIC RECOURSE

## ABSTRACT

Algorithmic Recourse (AR) aims to provide users with actionable steps to overturn unfavourable decisions made by machine learning predictors. However, these actions often take time to implement (*e.g.*, getting a degree can take years), and their effects may vary as the world evolves. Thus, it is natural to ask for recourse that remains valid in a dynamic environment. In this paper, we study the robustness of algorithmic recourse over time by casting the problem through the lens of causality. We demonstrate theoretically and empirically that (even robust) causal AR methods can fail over time except in the – unlikely – case that the world is *stationary*. Even more critically, unless the world is fully deterministic, *counterfactual* AR cannot be solved optimally. To account for this, we propose a simple yet effective algorithm for temporal AR that explicitly accounts for time under the assumption of having access to an estimator of the stochastic process. Our simulations on synthetic and realistic datasets show how considering time produces more resilient solutions to potential trends in the data distribution.

## 1 INTRODUCTION

Machine Learning (ML) models play an increasingly prominent role in high-stakes decision-making tasks like credit lending (Barbaglia et al., 2021), bail approval (Dressel & Farid, 2018) and medical diagnosis (Yoo et al., 2019). The general consensus is that, to ensure fairness, these systems need to provide users with tools to challenge their ruling, thus preserving human agency. These requirements are also being mandated by recent AI legislation (AI Act, 2021). **Algorithmic Recourse** (AR) (Karimi et al., 2022) aims to identify counterfactual explanations that users can follow to overturn unfavourable machine decisions. For instance, AR methods might suggest a user obtain a master's degree as this will net them a higher income and, in turn, higher chances of obtaining a loan.

In order to be of use, suggested recourse must be *actionable* (Ustun et al., 2019) and sufficiently inexpensive for the user to implement (De Toni et al., 2023b). We argue that actionability subsumes the notion of *timing*. Indeed, in practical applications, (*i*) recourse takes time to be implemented and to have an impact, (*ii*) performing the same action at different times might produce different effects. For instance, getting a degree takes years and only impacts salary after some time. Moreover, getting a degree at an older age reduces the expected salary increase. Our key insight is that, since ***time plays a key role in the effectiveness of recourse***, one has to ensure recourse suggestions should be *robust to time*, *i.e.*, they should lead to a positive outcome *irrespectively* of when the user performs them, or at least for a user-defined point in the future. See Fig. 1 for an illustration.

Many efforts have been directed at robustifying recourse suggestions in several scenarios (Jiang et al., 2024). For example, Upadhyay et al. (2021) studies recourse under model shift due to, *e.g.*, retraining, Pawelczyk et al. (2022a) addresses recourse in case the user's implementation of recourse is imperfect, and Dominguez-Olmedo et al. (2022) considers robustness to misspecification of the input instance. To the best of our knowledge, little has been done to explicitly formalize recourse robustness over time. Recently, Beretta & Cinquini (2023) evaluated the effect of time in AR by incorporating it in the recourse cost, thus not considering its effect on the validity.

Following Dominguez-Olmedo et al. (2022) and Beretta & Cinquini (2023), we study the problem of time in AR through the lens of *causality* (Pearl, 2009). In causal recourse, recourse suggestions are modelled as *interventions* on the user's features (Karimi et al., 2021), thus giving a reliable representation of how the features will change as the user acts on them to achieve recourse, provided we know the (approximate) causal model (Karimi et al., 2020b). We consider a novel setting in

which we are asked to provide recourse for a causal (non-stationary) discrete-time stochastic process subjected to trends. Our results challenge the usefulness of the mainstream variants of causal and non-causal AR extended in time, showing their recommendations ***can become invalid*** over time.

**Our contributions.** Summarizing, we (i) introduce a sound but intuitive formalization of temporal causal AR (Definition 1), based on causality (Pearl, 2009) and time-series with independent noise (Peters et al., 2013), (ii) show theoretically how uncertainty and non-stationarity hinder optimal counterfactual and sub-population recourse (CAR and SAR, (Karimi et al., 2020b)) for simple discrete-time stochastic processes, (iii) show how robustifying recourse via uncertainty sets is not enough to counteract time (Proposition 5), and (iv) present numerical simulations showcasing the detrimental effects of time on robust (non-)causal AR approaches, and showcase how a simple time-aware algorithm (Algorithm 1) can lessen such effects in synthetic and realistic settings (Section 4).

## 2 PRELIMINARIES AND RELATED WORK

Throughout, we indicate (random) variables $X$ in upper case, constants $x$ in lower case, vectors in bold $\mathbf{x}$, and sets $\mathcal{X}$ in italics. We also abbreviate $\{1, \ldots, n\}$ as $[n]$.

**Causality**. *Structural Causal Models* (SCMs) (Pearl, 2009) allow us to formalize and reason about the causal behaviour of a system. An SCM $\mathcal{M} = (\mathbf{X}, \mathbf{U}, P, \mathcal{F})$ encompasses endogenous variables $\mathbf{X} = \{X_i\}_{i=1}^d$, noise variables $\mathbf{U} = \{U_i\}_{i=1}^d$ distributed according to $P(\mathbf{U})$, and structural assignments $\mathcal{F}$ of the form $X_i := f_i(\mathbf{Pa}_i, U_i)$ that describe all causal relationships between variables and their *direct causes* (or parents) $\mathbf{Pa}_i \subseteq \mathbf{X} \setminus X_i$. An SCM induces a pushforward distribution $P(\mathbf{X}, \mathbf{U}) = P(\mathbf{X} \mid \mathbf{U})P(\mathbf{U})$, where $P(\mathbf{X} \mid \mathbf{U})$ is deterministic. *Hard interventions* $do(\mathbf{X}_{\mathcal{I}} = \boldsymbol{\theta})$ allow to implement external actions on an SCM. They

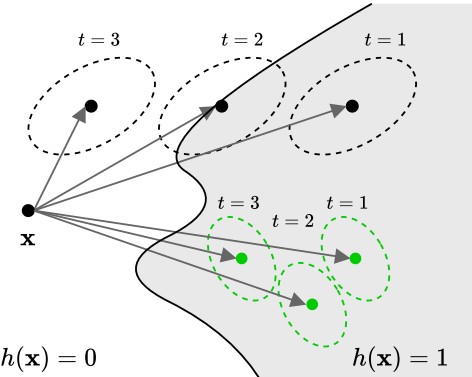

Figure 1: The validity of executed recourse suggestions can change over time $t$ (**black**) and AR methods should be robust to this effect (**green**).

replace a subset of variables $\mathbf{X}_{\mathcal{I}} \subseteq \mathbf{X}$ with constants $\boldsymbol{\theta} \in \mathbb{R}^{|\mathbf{X}_{\mathcal{I}}|}$, detached from their original parents, yielding a new SCM $\mathcal{M}^{do(\boldsymbol{\theta})}$ with updated structural assignments $\mathcal{F}^{do(\boldsymbol{\theta})}$ and an associated ***interventional distribution*** $P^{do(\boldsymbol{\theta})}(\mathbf{X})$. *Soft interventions* $do(\mathbf{X}_{\mathcal{I}} = \mathbf{x}_{\mathcal{I}} + \boldsymbol{\theta})$ change how the affected variables depend on their parents without detaching them. We shorten both kinds of intervention as $do(\boldsymbol{\theta})$, for simplicity. SCMs also enable us to reason *counterfactually* about what would have happened if the world were different due to an intervention $do(\boldsymbol{\theta})$, all else being equal. Given a realization $\mathbf{x}$, the ***counterfactual distribution*** $P^{do(\boldsymbol{\theta}), \mathbf{X}=\mathbf{x}}(\mathbf{X})$ is obtained by first *abducing* the exogenous factors $\mathbf{U}$ in the original SCM and then inferring the state of $\mathbf{X}$ in the intervened SCM, that is, $P^{do(\boldsymbol{\theta}), \mathbf{X}=\mathbf{x}}(\mathbf{X}) = P^{do(\boldsymbol{\theta})}(\mathbf{X} \mid \mathbf{U})P(\mathbf{U} \mid \mathbf{X} = \mathbf{x})$ (Pearl, 2009, Theorem 7.1.7). If the structural equations are invertible, $P(\mathbf{U} \mid \mathbf{x})$ is deterministic, and so is the counterfactual distribution.

**Causal Algorithmic Recourse**. In AR, the main quantities of interest are the *user's state* $\mathbf{x} \sim P(\mathbf{X})$ and the *outcome* $y \sim P(Y \mid \mathbf{X})$, *e.g.*, the event that the user will repay their loan. The SCM underlying $P(\mathbf{X})$ is assumed to be known or estimated from data (Karimi et al., 2020b), enabling us to apply interventions to evaluate the effect of changing the user's state while considering all causal dependencies between variables. Given a (potentially black-box) classifier $h : \mathbf{x} \mapsto [0, 1]$ approximating $P(Y \mid \mathbf{X})$ and a realization $\mathbf{x}$ yielding an undesirable outcome, *i.e.*, $h(\mathbf{x}) < 1/2$, AR involves finding an intervention $\boldsymbol{\theta}^*$ that, once implemented by the user, leads in expectation to a more favourable outcome. There are two mainstream approaches to causal AR (Karimi et al., 2020b). ***Sub-population recourse*** (SAR) provides recourse to users belonging to a specific sub-group, and it is defined as:[1]

$$\boldsymbol{\theta}^* \in \operatorname{argmin}_{\boldsymbol{\theta} \in \mathbb{R}^d} \ \mathbb{E}_{\hat{\mathbf{x}} \sim P^{do(\mathbf{X}_{\mathcal{I}} = \boldsymbol{\theta})}(\mathbf{X})}[C(\hat{\mathbf{x}}, \mathbf{x})] \quad \text{s.t.} \quad \mathbb{E}_{\hat{\mathbf{x}} \sim P^{do(\mathbf{X}_{\mathcal{I}} = \boldsymbol{\theta})}(\mathbf{X})}[h(\hat{\mathbf{x}})] \geq 1/2 \quad (1)$$

---

[1]The choice of $\boldsymbol{\theta}$ is often restricted by actionability requirements (*e.g.*, age cannot be changed at will) or other constraints (*e.g.*, monotonicity: age can only increase). We omit this detail for readability.

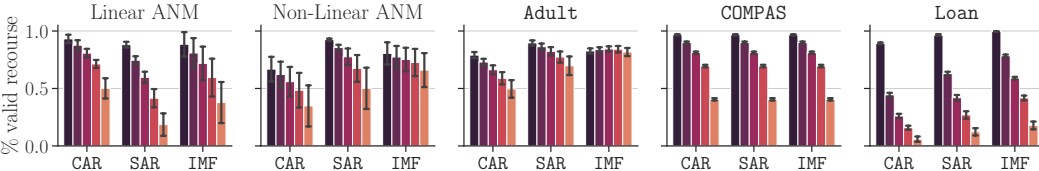

Figure 2: **Algorithmic recourse is not robust in time.** Empirical average validity and standard deviation of robust counterfactual (CAR), sub-population (SAR) and non-causal recourse (IMF) at time $t = 50$ for synthetic (Section 4.1) and realistic (Section 4.2) time series with a non-linear trend. We report the validity varying the strength $\alpha$ of the trend. Legend: ■ 0 ■ 0.3 ■ 0.5 ■ 0.7 ■ 1.0.

Here, $C$ is a non-negative cost function measuring the user's effort, *e.g.*, the $\ell_2$-norm. Selecting a specific subgroup amounts to conditioning $P^{do(\mathbf{X}=\theta)}$ in Eq. (1) on a subset of variables $\mathbf{X}_{nd(\mathcal{I})} = \mathbf{x}_{nd(\mathcal{I})}$ where $nd(\mathcal{I})$ indicates the non-descendants of the intervened upon variables $\mathcal{I}$. ***Counterfactual recourse*** (CAR) allows computing *individualized* recourse for a specific individual, identified by $\mathbf{x}$. It is formulated like Eq. (1), except that the interventional distribution $P^{do(\theta)}(\mathbf{X})$ is replaced with the counterfactual distribution $P^{do(\theta),\mathbf{X}=\mathbf{x}}(\mathbf{X})$. Lastly, since providing optimal recourse (Eq. (1)) is harder for unrestricted SCMs (Karimi et al., 2020b), causal AR typically assumes the SCM of $P(\mathbf{X})$ belongs to an identifiable and invertible class, *e.g.*, Additive Noise Models (ANMs) (Karimi et al., 2021; Dominguez-Olmedo et al., 2022; Karimi et al., 2020a).

**Further related works.** Our work is related to works on robust AR, counterfactual explanations for time series and causality. The literature on the robustness of algorithmic recourse aims at generating counterfactual explanations which are robust to the type of model $h$ updates and changes (Ferrario & Loi, 2022; Pawelczyk et al., 2022b; Nguyen et al., 2023; Meyer et al., 2023; Upadhyay et al., 2021), endogenous dynamics (Altmeyer et al., 2023), model multiplicity (Pawelczyk et al., 2020; Leofante et al., 2023), noisy execution of the intervention (Pawelczyk et al., 2022a; Virgolin & Fracaros, 2023) or uncertainty on the input instance (Dominguez-Olmedo et al., 2022; Slack et al., 2021; Artelt et al., 2021). See Jiang et al. (2024) for a recent comprehensive survey on the topic. None of these works explicitly formalize time as a dimension in their computational model. Recently, Fonseca et al. (2023) studied empirically algorithmic recourse in a multi-agent setting where different users compete for resources over time, without a causal notion. The literature on counterfactual explanations for multivariate time series provides instead techniques to generate explanations for stochastic processes (Delaney et al., 2021; Ates et al., 2021) by assuming either *independent manipulable features* (IMF) (*i.e.*, no causal relationships between variables) or a simpler form of causality *e.g.*, Granger causality (Granger, 1969).

## 3 ALGORITHMIC RECOURSE IN TIME

In real-world applications, users do not implement nor complete suggested interventions immediately. For instance, obtaining a degree can take years. This is problematic because $P(\mathbf{X}, Y)$ might change over time due to, *e.g.*, inflation rates, seasonality of loan interests and classifier updates, respectively, meaning that *recourse produced by existing approaches could become ineffective or even counterproductive in the future*. Fig. 2 shows the empirical average validity (% of interventions achieving recourse) of state-of-the-art robust (non-)causal recourse methods on different binary decision problems, where $P(\mathbf{X}, Y)$ is a stochastic process exhibiting a non-linear trend. Unfortunately, *current robust (non-)causal recourse methods are increasingly fragile to time proportionally to the trend's strength*. This section is devoted to this issue. We summarize our theoretical results and the characteristics of the methods for (non-)causal recourse in Table 1 in Appendix A.

### 3.1 FORMALIZING TEMPORAL CAUSAL AR

Before proceeding, we need to extend causal AR with a time dimension. We do so by considering a stochastic process $P(\mathbf{X}^t, Y^t)$ capturing the evolution of the user's state and its relationship to the target variable over time $t \in \mathbb{N}$. In the following, we assume that $P$ is induced by an SCM over *the same variables* in which the parents of each variable lie in the past, and specifically within a fixed (but otherwise arbitrary) horizon $\rho \geq 1$, *i.e.*, $\mathbf{Pa}_{X_i^t} \subseteq \bigcup_{\delta=0}^{\rho} \mathbf{X}^{t-\delta}$, and similarly for $Y^t$. We

also assume that the outcome $Y^t$ only depends on the user's states $\mathbf{X}^1, \ldots, \mathbf{X}^t$ up to time $t$. We investigate the effect of time on recourse by considering stochastic processes where $P(\mathbf{X}^t)$ might not be stationary, while the conditional distribution $P(Y^t \mid \mathbf{X}^t)$ is unchanged, similarly to *covariate shift* (Shimodaira, 2000). For this reason, we can assume the classifier $h^t$ is fixed.[2] This allows us to formulate Temporal Causal AR as follows:

**Definition 1** (Temporal Causal AR). *Consider a stochastic process* $P(\mathbf{X}^t, Y^t)$, *a cost function* $C(\cdot, \cdot)$ *(e.g., the* $L_2$ *norm), a constant classifier* $h$ *and a user* $\mathbf{x}^t$ *such that* $h(\mathbf{x}^t) < 1/2$. *Assume the user will perform the intervention at a later time* $t + \tau$, *for a fixed* $\tau > 0$. *We want to find the cheapest intervention* $do(\mathbf{X}_{\mathcal{I}}^{t+\tau} = \mathbf{x}_{\mathcal{I}}^{t+\tau} + \boldsymbol{\theta})$ *that achieve recourse, in expectation, when applied at time* $t + \tau$:

$$
\boldsymbol{\theta}^* \in \min_{\substack{\boldsymbol{\theta} \in \mathbb{R}^d}} \mathbb{E}_{\substack{\hat{\mathbf{x}}^{t+\tau} \sim Q(\mathbf{X}^{t+\tau}; \mathbf{x}^{t+\tau}, \boldsymbol{\theta}) \\ \mathbf{x}^{t+\tau} \sim P(\mathbf{X}^{t+\tau} | \mathbf{X}^t = \mathbf{x}^t)}}[C(\hat{\mathbf{x}}^{t+\tau}, \mathbf{x}^{t+\tau})] \quad \text{s.t.} \quad \mathbb{E}_{\substack{\hat{\mathbf{x}}^{t+\tau} \sim Q(\mathbf{X}^{t+\tau}; \mathbf{x}^{t+\tau}, \boldsymbol{\theta}) \\ \mathbf{x}^{t+\tau} \sim P(\mathbf{X}^{t+\tau} | \mathbf{X}^t = \mathbf{x}^t)}}\left[h(\hat{\mathbf{x}}^{t+\tau})\right] \geq 1/2
$$

(2)

*where* $Q(\mathbf{X}^{t+\tau}; \mathbf{x}^t, \boldsymbol{\theta})$ *can be the interventional distribution* $P^{do(\mathbf{X}_{\mathcal{I}}^{t+\tau} = \mathbf{x}_{\mathcal{I}}^{t+\tau} + \boldsymbol{\theta})}(\mathbf{X}^{t+\tau} \mid \mathbf{X}_{nd(\mathcal{I})} = \mathbf{x}_{nd(\mathcal{I})})$, *or the counterfactual distribution* $P^{do(\mathbf{X}_{\mathcal{I}}^{t+\tau} = \mathbf{x}_{\mathcal{I}}^{t+\tau} + \boldsymbol{\theta}); \mathbf{X}^{t+\tau} = \mathbf{x}^{t+\tau}}(\mathbf{X}^{t+\tau})$.

Definition 1 deserves some discussion. First of all, it describes both ***temporal subpopulation causal AR*** (`T-SAR`) and ***temporal counterfactual causal AR*** (`T-CAR`). We remark that, while practical solutions for `T-SAR` can be devised (see Section 3.6), `T-CAR` is intrinsically more challenging (as we discuss in Section 3.2). Additionally, this formulation assumes that recourse is implemented, and its causal effects are observed, at time $t + \tau$, for a fixed $\tau$, thus our SCM must exhibit *instantaneous effects* (Peters et al., 2013).

**Causal time series models**. In the remainder we assume the stochastic process is a *Time series Model with Independent Noise* (TiMINo), adapted from (Peters et al., 2013):

**Definition 2** (TiMINo for Algorithmic Recourse). $P(\mathbf{X}^t, Y^t)$ *satisfies TiMINo if it causally factorizes as* $X_i^t = f_{X_i}(\mathbf{Pa}_{X_i^t}) + U_{X_i}^t$ *and* $Y^t = f_Y^t(\mathbf{X}^t) + U_Y^t$, *for all* $i \in [d]$, *where* $U_{X_i}^t, U_Y^t$ *are jointly independent and identically distributed for all* $i \in [d]$ *and* $t \in \mathbb{N}$.

Under appropriate conditions, TiMINo SCMs are *invertible*, allowing us to apply causal reasoning to infer the counterfactual distribution when computing AR, and can be identified from observational data. Specifically, under appropriate choices of the family of $f_{X_i}^t$, $f_Y^t$ – which still allow them to be non-linear – and $P(\mathbf{U}^t)$ we are guaranteed to identify both the *summary graph* and *full-time* graph (Peters et al., 2013). Moreover, Peters et al. (2013) provide a causal discovery procedure for *TiMINo* time series that avoids drawing wrong causal conclusions in the presence of confounders. Throughout the paper, we assume the full-time graph is *sufficient* (*e.g.*, there are no unobserved confounders (Peters et al., 2017)). We will show how, even in this optimistic scenario, our negative results for temporal recourse still hold.

**Interventions in time.** Intuitively, time can only invalidate recourse as long as those changes occurring after the time $t$ at which recourse is issued can influence the distribution of the future state $\mathbf{X}^{t+\tau}$. In the unlikely case that recourse is a hard intervention $do(\mathbf{X}^{t+\tau} = \boldsymbol{\theta})$ affecting *all* variables $\mathbf{X}$, then $\mathbf{X}^{t+\tau}$ no longer depends on its past states because such interventions *detach* all variables from their parents, overriding possible temporal effects (Dominguez-Olmedo et al., 2022). Hence, recourse remains valid *by construction*. In practice, however, recourse *i)* is often *restricted to few variables*, due to, *e.g.*, actionability and cost constraints, or *ii)* may be defined as a *soft intervention*, that is, an intervention of the form $do(\mathbf{X}_{\mathcal{I}}^t = \mathbf{x}_{\mathcal{I}}^t + \boldsymbol{\theta})$, that does *not* detach $\mathbf{X}^t$ (Dominguez-Olmedo et al., 2022). In the following, we focus on this more realistic setting with soft interventions. For the rest of the paper, in the context of temporal algorithmic recourse, we will use the $do(\boldsymbol{\theta})$ notation to represent an intervention **always** applied at time $t + \tau$, unless specified otherwise.

**On the naïve solution.** One obvious "remedy" to counter the impact of time is to simply allow users to re-compute recourse at time $t + \tau$, using the new user's state $\mathbf{x}^{t+\tau}$. However, implementing this new suggestion might itself take time $\tau' > \tau$, meaning this does not prevent invalidation at all.

---

[2] If this is not the case, one option is to leverage existing techniques for addressing changes due to retraining, such as Upadhyay et al. (2021) and Pawelczyk et al. (2022b). These works and ours are complementary and studying their interplay is a promising avenue for future work.

## 3.2 Uncertainty over time compromises counterfactual recourse

We begin by studying temporal counterfactual AR (T-CAR). It provides *invididualized* suggestions, thus achieving the true optimal intervention for a given user (Karimi et al., 2020b). According to Definition 1, it must be computed considering the *future* counterfactual distribution $P^{do(\boldsymbol{\theta});\mathbf{X}^{t+\tau}=\mathbf{x}^{t+\tau}}(\mathbf{X}^{t+\tau})$. Unfortunately, this turns out to be problematic. The following proposition shows that this distribution cannot be recovered exactly except under strong assumptions.

**Proposition 1.** *Let $P(\mathbf{X}^t, Y^t)$ satisfy TiMINo. Given a realization $\mathbf{x}^t$ and an intervention $\boldsymbol{\theta} \in \mathbb{R}^d$, we can recover the counterfactual distribution over the future $P^{do(\boldsymbol{\theta}),\mathbf{X}^t=\mathbf{x}^t}(\mathbf{X}^{t+\tau})$ if and only if, for all $t > 0$, $\mathrm{Var}(\mathbf{U}^t) = 0$ and $\mathbb{E}[\mathbf{U}^t]$ are constant.*

All proofs can be found in Appendix B. In words, given $\mathbf{x}^t$, one cannot know the true counterfactual distribution at time $t + \tau$ unless all exogenous factors have zero variance, that is, $P(\mathbf{X}^{t+\tau}, \ldots, \mathbf{X}^{t+1} \mid \mathbf{x}^t)$ is *deterministic*. Proposition 1 has profound consequences for recourse because, recalling the central role of the counterfactual distribution in Eq. (1), it entails that **for non-deterministic processes we cannot solve T-CAR optimally**.

**Corollary 2** (Informal). *Let $P(\mathbf{X}^t)$ satisfy TiMINo and consider a constant injective classifier $h$. Given a realization $\mathbf{x}^t$, a counterfactual recourse $\boldsymbol{\theta} \in \mathbb{R}^d$ applied at time $t + \tau$, with $\tau > 0$, cannot be optimal unless exogenous factors have zero variance.*

We explore this issue empirically in Section 4. We remark that Proposition 1 also holds for non-TiMINo stochastic processes as long as they admit performing abduction (*e.g.*, the structural equations are invertible), and so does Corollary 2.

## 3.3 Sub-population AR deteriorates in a non-stationary world

Given the inherent limitations of temporal counterfactual AR, in the remainder, we focus on temporal *sub-population* AR (T-SAR), which is generally regarded as the most *plausible* form of recourse (Karimi et al., 2020b). The next proposition shows that, insofar as $P(\mathbf{X}^t, Y^t)$ is *stationary*[3] and the classifier $h$ is constant and injective, recourse that is optimal for *static* sub-population recourse (Eq. (1)) remains optimal over time (Eq. (2)).

**Proposition 3.** *Consider a stationary stochastic process $P(\mathbf{X}^t)$ and a constant injective classifier $h$. Any optimum $\boldsymbol{\theta}^*$ of Eq. (1) is also optimal for Eq. (2) for any time lag $\tau \in \mathbb{N}$.*

Despite this positive result, the issue is that **stationarity is seldom satisfied in practice**: many real-world processes exhibit trends (*e.g.*, inflation rate, seasonality of loan interests, *etc.*). The next example shows how recourse can become invalid if the $P(\mathbf{X}^t, Y^t)$ is *not* stationary, even for a simple one-dimensional *trend-stationary*[4] stochastic process. Full derivations are in Appendix B.

**Example 1.** *Consider a trend-stationary stochastic process defined by these structural equations:*

$$X^t = \alpha X^{t-1} + m(t) + U_X^t, \quad U_X^t \sim \mathcal{N}(\mu_X, \sigma_X)$$
$$Y^t = \beta X^t + U_Y^t, \qquad\qquad U_Y^t \sim \mathcal{N}(0, 1) \tag{3}$$

*for all $t$, where $\alpha \in (0, 1)$ and $\beta \in \mathbb{R}$. The function $m(t) : \mathbb{R} \to \mathbb{R}$ represents a trend independent of $X^t$ and $Y^t$. We consider a linear trend $m(t) = -ct + U_m^t$, where $U_m^t \sim \mathcal{N}(\mu_m, \sigma_m)$ and $c \in \mathbb{R}^+$. Consider the fixed classifier $h(X^t) = \sigma(Y^t \mid X^t)$ where $\sigma(x) = 1/(1 + e^{-x})$. We have that the optimal intervention $\theta^{t+\tau} \in \mathbb{R}$ for which we have $\mathbb{E}[h(X^{t+\tau} + \theta)] \geq 1/2$ can be expressed as:*

$$\theta^{t+\tau} = -\alpha^{\tau+1} x^{t-1} - \sum_{i=0}^{\tau} \alpha^{\tau-i}(-c(t+i) + \mu_m + \mu_X) \tag{4}$$

Since $m(t)$ is monotonically decreasing for $c > 0$, we have the optimal interventions satisfy $\theta^t \leq \theta^{t+\tau}$, implying that **a recourse issued at time $t$ becomes invalid as time passes**. Following Proposition 3, we can state the following general corollary regarding our ability to provide optimal subpopulation recourse for general stochastic processes:

---

[3]A discrete stochastic process $\{\mathbf{X}^t\}_{t\in\mathbb{N}}$ is (weak-sense) stationary when it satisfies the following properties: $\mathbb{E}[\mathbf{X}^{t+\tau} - \mathbf{X}^t] = 0$ and $K(t + \tau, t) = K(\tau, 0)$ for all $t, \tau \in \mathbb{N}$ where $K(p, q) = \mathbb{E}\left[(X^p - \mathbb{E}[X^p])(X^q - \mathbb{E}[X^q])\right]$ is the autocovariance and $\mathbb{E}[|\mathbf{X}^t|^2] < \infty$ for all $t \in \mathbb{N}$.

[4]A stochastic process $\{\mathbf{X}^t\}_{t\in\mathbb{N}}$ is trend-stationary when it can be expressed as $\mathbf{X}^t = f(t) + \mathbf{e}^t$, where $f(t)$ is (non-)linear trend function and $\mathbf{e}^t$ is a stationary stochastic process.

**Corollary 4** (Informal). *Consider a discrete-time process $P(\mathbf{X}^t)$ and a constant injective classifier $h$. Unless $P(\mathbf{X}^t)$ is stationary, the optimal intervention $\boldsymbol{\theta}^*$ achieving recourse can vary depending on $t, \tau \in \mathbb{N}$.*

### 3.4 ROBUST ALGORITHMIC RECOURSE IS NOT ENOUGH TO COUNTERACT TIME

Given the previous results, we could imagine to *robustify* the recourse procedure to account for non-stationarity of $P(\mathbf{X}^t, Y^t)$. For example, a common solution to robustify CAR and SAR is to provide an intervention $\boldsymbol{\theta}$ achieving recourse within a causal "uncertainty set" $B(\mathbf{X}; \boldsymbol{\Delta})$, defined below (Dominguez-Olmedo et al., 2022). In this section, we show how *set-based robust causal recourse method falls short when dealing with time*. We first start by defining robust causal AR:

**Definition 3** (Adapted from Dominguez-Olmedo et al. (2022)). *Consider a realization $\mathbf{x} \in \mathbb{R}^d$, a norm $|| \cdot ||$, and a tolerance $\epsilon > 0$. We define a causal uncertainty set $B(\mathbf{x}; \boldsymbol{\Delta}) = \{\mathbf{x}' \sim P^{do(\boldsymbol{\Delta}), \mathbf{X}=\mathbf{x}}(\mathbf{X}) : ||\boldsymbol{\Delta}|| \leq \epsilon\}$ as the collection of the causal counterfactuals under small additive perturbations $\boldsymbol{\Delta} \in \mathbb{R}^d$. We want to find the cost-minimizing intervention $\boldsymbol{\theta}$ achieving recourse in all the region defined by $B(\mathbf{x}; \boldsymbol{\Delta})$. Thus, the optimization objective for robust recourse becomes:*

$$\boldsymbol{\theta}^* \in \operatorname*{argmin}_{\boldsymbol{\theta} \in \mathbb{R}^d} \mathbb{E}[C(\mathbf{x}, \hat{\mathbf{x}})] \quad \text{s.t.} \quad \mathbb{E}[h(\hat{\mathbf{x}})] \geq 1/2 \quad \forall \ \hat{\mathbf{x}} \sim B(\mathbf{x}; \boldsymbol{\Delta}) \tag{5}$$

*where $\hat{\mathbf{x}}$ is distributed according to either the counterfactual or interventional distribution.*

A robust intervention might still obtain recourse for later time steps depending on the tolerance $\epsilon$. Intuitively, by asking the user to perform a more difficult action (*e.g.*, increase your income by \$1000, instead of \$100), we can provide interventions that are less susceptible to potential dynamics. However, if the intervention is applied too late, we will not achieve recourse:

**Proposition 5.** *Consider a fixed $\epsilon > 0$, a trend-stationary process $P(\mathbf{X}^t)$, a constant injective classifier $h$ and realization $\mathbf{x}^t$ where $h(\mathbf{x}^t) < 1/2$. Let us assume we have an optimal $\epsilon$-robust intervention $\boldsymbol{\theta}$ for timestep $t$. There always exists a trend $m : \mathbb{N} \to \mathbb{R}$ and a positive $\tau$, such that $\mathbb{E}_{\mathbf{x}^{t+\tau} \sim P^{do(\boldsymbol{\theta})}(\mathbf{X}^{t+\tau} | \mathbf{X}^{t+\tau}_{nd(\mathcal{I})} = \mathbf{x}^{t+\tau}_{nd(\mathcal{I})}, \mathbf{X}^t = \mathbf{x}^t)}[h(\mathbf{x}^{t+\tau})] < 1/2$.*

### 3.5 ON THE STABILITY OF RECOURSE OVER TIME

In Sections 3.2 to 3.4, we showed how recourse validity can be compromised by the uncertainty and non-stationarity of the stochastic process $P(\mathbf{X}^t, Y^t)$, and we also showed how set-based robustness techniques fail over time. However, users might be willing to accept recourses that slowly become less effective rather than performing more challenging interventions. Thus, we now characterize instead the *rate* at which our recourse suggestion validity decreases. Additionally, we assume a non-stationary $P(Y^t | \mathbf{X}^t)$ approximated by a sequence of classifiers $h^t$, one for each $t \in \mathbb{N}$.

**Definition 4** (Temporal recourse invalidation rate). *Consider a discrete-time stochastic process $P(\mathbf{X}^t, Y^t)$, and any classifier $h^t$ approximating $P(Y^t | \mathbf{X}^t)$ for each $t \in \mathbb{N}$. Given a realization $\mathbf{x}^t$ and an intervention $\boldsymbol{\theta}$ such that $\mathbb{E}[h(\hat{\mathbf{x}}^t)] \geq 1/2$, where $\hat{\mathbf{x}}^t \sim P^{do(\boldsymbol{\theta})}(\mathbf{X}^t | \mathbf{X}^t_{nd(\mathcal{I})} = \mathbf{x}^t_{nd(\mathcal{I})})$, we define the temporal invalidation rate after a time-lag $\tau > 0$ as:*

$$\Delta h(\boldsymbol{\theta}; \tau) = \mathbb{E}\left[\left|h^{t+\tau}(\hat{\mathbf{x}}^{t+\tau}) - h^t(\hat{\mathbf{x}}^t)\right|\right] \tag{6}$$

*where $\hat{\mathbf{x}}^{t+\tau} \sim P^{do(\boldsymbol{\theta})}(\mathbf{X}^{t+\tau} | \mathbf{X}^{t+\tau}_{nd(\mathcal{I})} = \mathbf{x}^{t+\tau}_{nd(\mathcal{I})})$.*

Let us now consider the setting in which we have a bounded stochastic process $P(\mathbf{X}^t, Y^t)$, where $-k \leq X_i^t \leq k$ for some $k \in \mathbb{R}^+$ for all $t \in \mathbb{N}$. If we have access to a dataset $\mathcal{D}^t \sim P(\mathbf{X}^t, Y^t)$ sampled from the stochastic process, we can use it to train a classifier $h^t$ via empirical risk minimization. Let us consider a *linear* classifier $h^t(\mathbf{x}) = \langle \boldsymbol{\beta}^t, \mathbf{x}^t \rangle$ with bounded weights $-k \leq \beta_i^t \leq k$ *e.g.*, trained via Bounded Least-Squares (BLS) (Stark & Parker, 1995). Then, given an intervention $\boldsymbol{\theta}$, we can derive the following upper bound on the recourse instability within a time interval $(t, t + \tau)$.

**Theorem 6** (Upper-bound invalidation rate). *Consider a discrete-time stochastic process $P(\mathbf{X}^t, Y^t)$ and a sequence of linear classifiers $h^t(\mathbf{x}^t) = \langle \boldsymbol{\beta}^t, \mathbf{x}^t \rangle$ approximating $P(Y^t | \mathbf{X}^t)$. We assume $-k \leq \beta_i, X_i \leq k$ for $k \in \mathbb{R}^+$. The temporal invalidation rate is upper bounded as follows:*

$$\Delta h(\boldsymbol{\theta}; \tau) \leq k\sqrt{d} \cdot \mathbb{E}\left[\|\boldsymbol{\beta}^{t+\tau} - \boldsymbol{\beta}^t\|\right] + \mathbb{E}\left[\|\hat{\mathbf{x}}^{t+\tau} - \hat{\mathbf{x}}^t\|\right] \tag{7}$$

*where $\|\cdot\|$ is the $\ell_2$-norm and the expectation is over $\mathcal{D}^t \sim P(\mathbf{X}^t, Y^t)$ and the training process.*

---

**Algorithm 1** Generate robust recourse solutions for T-SAR given a future time-lag $\tau$, a differentiable classifier $h$, an estimator $\tilde{P}(\mathbf{X}^t)$ and a subset of intervened nodes $\mathcal{I} \subseteq [d]$.

---

**Require:** $\mathbf{x}^t$, individual; $N > 0$; $\lambda > 0$; $\eta > 0$;
1: $B(\mathbf{x}^t; \tau) \leftarrow \{\mathbf{x}' \sim \tilde{P}(\mathbf{X}^{t+\tau} \mid \mathbf{X}^t = \mathbf{x}^t)\}$
2: **for** $epochs = 1$ to $N$ **do**
3:      **while** $\exists \, \mathbf{x}' \in B(\mathbf{x}^t; \tau)$ such that $\mathrm{ER}(\mathbf{x}', \tau; \boldsymbol{\theta}) < 1/2$ **do**
4:          $\mathbf{x}^* \leftarrow \mathrm{argmin}_{\mathbf{x}' \in B(\mathbf{x}^t; \tau)} \mathrm{ER}(\mathbf{x}', \tau; \boldsymbol{\theta})$
5:          $\mathcal{L} \leftarrow \|\boldsymbol{\theta}\| - \lambda(\mathrm{ER}(\mathbf{x}^*, \tau; \boldsymbol{\theta}) - 1/2)$
6:          $\boldsymbol{\theta} \leftarrow \boldsymbol{\theta} - \eta \nabla \mathcal{L}$
7: **return** $\boldsymbol{\theta}$

---

Theorem 6 shows how the recourse instability is upper bounded by how much the world varies between $t$ and $t + \tau$ in terms of the data distribution $P(\mathbf{X}^t)$, and the classifier. Moreover, the size of the problem $d$ concurs by increasing the worst-case error at a sublinear rate. The upper bound can be useful for non-linear classifiers $h$ if we consider a linear function approximating locally (Simonyan et al., 2013) their decision function close to a realization $\mathbf{x}^t$, such as LIME (Ribeiro et al., 2016). If our stochastic process is *trend-stationary* as in Example 1, we can derive the following upper bound:

**Corollary 7.** *Consider a discrete-time trend-stationary stochastic process* $P(\mathbf{X}^t)$, *where* $m_i :$ $\mathbb{N} \to \mathbb{R}$ *represents the trend for* $X_i$ *and the classifiers* $h^t(\mathbf{x}^t) = \langle \boldsymbol{\beta}^t, \mathbf{x}^t \rangle$. *Let us define* $m^*(t) = \max_{i \in [d]} m_i(t)$ *as the largest trend for* $t \in \mathbb{N}$. *Then, we have the upper bound:*

$$\Delta h(\boldsymbol{\theta}; \tau) \leq k \left( \sqrt{d} \cdot \mathbb{E} \left[ \|\boldsymbol{\beta}^{t+\tau} - \boldsymbol{\beta}^t\| \right] + d \cdot (m^*(t + \tau) - m^*(t)) \right) \tag{8}$$

These results assume that the cost function remains constant over time. In Appendix H, we show an analogous result when this is not the case, as in *personalized cost functions* (De Toni et al., 2023b).

## 3.6 Accounting for Time in Practice

In Sections 3.2 to 3.5, we showed how recourse validity is hindered by the uncertainty and non-stationarity of the stochastic process. Given a factual instance $\mathbf{x}$ and a recourse $\boldsymbol{\theta}$, Theorem 6 also implies the upper bound can grow quickly depending on the difference of the induced interventional distributions and classifier $h$ between $t$ and $t + \tau$. Unfortunately, for any model $h$ and tolerance $\epsilon$, there can exist *multiple* (robust) recourses $\boldsymbol{\theta}$ to choose from (Pawelczyk et al., 2020). Since (robust) CAR and SAR have *no means to differentiate between recourses*, they might end up suggesting interventions which rapidly become invalid. An alternative is to settle for classical robust AR, which *can* provide some amount of safety w.r.t. time depending on the chosen epsilon $\epsilon$ (cf. Section 4). The issue with this is that, as shown by Proposition 5, choosing $\epsilon$ without considering how the world changes can be dramatically suboptimal, *i.e.*, robust AR might recommend expensive actions that risk becoming invalid.

Luckily, for SAR, we can mitigate these issues, as long as we have access to an estimator of the stochastic process. We present a simple algorithm (Algorithm 1) for temporal *sub-population* algorithmic recourse (Eq. (2)) drawing inspiration from *adversarially robust recourse* (Dominguez-Olmedo et al., 2022). Following the results of Section 3.4, we argue that, instead of providing robust recourse for an arbitrary uncertainty set with a fixed $\epsilon$, we need to provide a robust $\boldsymbol{\theta}$ for a *forecasted* region of the feature space. We do so by extending the notion of uncertainty set $B(\mathbf{x}^t; \tau)$ to consider the distribution entailed by the TiMINo SCM conditioned on the observed realization, $P(\mathbf{X}^{t+\tau} \mid \mathbf{X}^t = \mathbf{x}^t)$, after a time lag $\tau > 0$:

$$B(\mathbf{x}^t; \tau) = \{\mathbf{x}' \sim P(\mathbf{X}^{t+\tau} \mid \mathbf{X}^t = \mathbf{x}^t)\} \tag{9}$$

Such a region does not depend on a fixed $\epsilon$, thus sidestepping the issue shown by Proposition 5. Algorithm 1 assumes to have access to a constant and *differentiable* classifier $h$, and to an estimator $\tilde{P}(\mathbf{X}^t)$ of the stochastic process. We define $\mathrm{ER}(\mathbf{x}, \tau; \boldsymbol{\theta}) = \mathbb{E}[h(\hat{\mathbf{x}})]$ where $\hat{\mathbf{x}}$ is sampled from the interventional distribution conditioned on the non-descendant $nd(\mathcal{I})$ of the intervened upon nodes $\mathcal{I}$. Similarly to Dominguez-Olmedo et al. (2022), in practice, we approximate $B(\mathbf{x}^t; \tau)$ by sampling a finite number of instances from $P(\mathbf{X}^{t+\tau} \mid \mathbf{X}^t = \mathbf{x}^t)$. As usual, users can control the trade-off between cost and robustness by varying $\lambda$ within the Lagrangian (line 5, Algorithm 1).

**Computational complexity.** The running time of Algorithm 1 depends on (i) the number of epochs $N$, and (ii) an upper bound on the number of iterations $K$ of the inner loop (lines 3-6). Considering all potential variable subsets $\mathcal{I} \subset [d]$, the complexity is $O(NK2^d)$. Luckily, not all features are actionable, and the general wisdom is to provide sparse solutions (*e.g.*, by considering only $|\mathcal{I}| \leq m$ sets) so we have $O(NK\binom{d}{m})$ if $m \ll d$. Lastly, the time-lag $\tau$ has no impact on the running time of the algorithm, but in practice, we would need to run Algorithm 1 for each $\tau$ specified by the user.

**Relationship between Algorithm 1 and other causal methods.** Algorithm 1 subsumes existing causal recourse methods. If we replace $B(\mathbf{x}^t; \tau)$ with the uncertainty set in Definition 3, we obtain the *robust* counterfactual (CAR) or sub-population (SAR) recourse method (Dominguez-Olmedo et al., 2022), depending on how we define the distribution over the expectation in $\mathrm{ER}(\mathbf{x}, \tau; \boldsymbol{\theta})$. Moreover, if we do not consider $\tau$ such as $\mathrm{ER}(\mathbf{x}; \boldsymbol{\theta}) = h(\mathbf{x} + \boldsymbol{\theta})$ and $B(\mathbf{x}^t; \boldsymbol{\Delta}) = \{\mathbf{x}^t + \boldsymbol{\Delta} : ||\boldsymbol{\Delta}|| \leq \epsilon\}$, we obtain (robust) non-causal recourse (IMF, Wachter et al. (2017)).

## 4  EXPERIMENTS AND RESULTS

In this section, we empirically study the effect of time on recourse validity in synthetic and realistic settings taken from the literature, by comparing Algorithm 1 against several robust (non-)causal AR methods. See Appendix C for a detailed explanation of the experimental setting and techniques.

### 4.1  EXPERIMENTS WITH SYNTHETIC TIME-SERIES

**Experimental setup.** First, we consider the linear and non-linear 3-variable synthetic ANMs from Karimi et al. (2021) representing a binary decision problem (*e.g.*, loan granted/denied). We adapt them to describe a trend-stationary stochastic process by adding an additive trend function $m(t) = \alpha \cdot (\beta_l \cdot l(t) + \beta_s \cdot s(t))$ to the structural equations, where $l(t)$ and $s(t)$ are the *linear* and *seasonal* components. The parameter $\alpha \in (0, 1)$ governs the strength of the trend. We consider three types of trends: *linear* ($\beta_l > 0, \beta_s = 0$), *seasonal* ($\beta_l = 0, \beta_s > 0$) and *linear+seasonal* ($\beta_l > 0, \beta_s > 0$). Then, we sample a time series for each ANM with 10000 individuals for $t \in [0, 100]$ timesteps. We split the time series into training (8000) and testing (2000) and train a fixed 3-layer MLP to approximate $P(Y^t \mid \mathbf{X}^t)$ using *only the training data at time $t = 0$*. We pick 250 individuals negatively classified ($h(\mathbf{x}) < 1/2$) by the MLP from the test set at time $t = 0$, and we compute recourse suggestions, by considering the $\ell_1$-norm as a cost function, with *robust* counterfactual and sub-population recourse (CAR and SAR, (Dominguez-Olmedo et al., 2022)), *robust* non-causal recourse (IMF, (Wachter et al., 2017)) and time-aware sub-population recourse (T-SAR, Algorithm 1).

We empirically choose a smaller and larger $\epsilon \in \{3, 5\}$ maximizing the robust methods' validity at $t = 0$. To simulate the user implementing the suggested intervention at a later time, we vary the time lag $\tau$ and compute the ***empirical average validity*** (% of interventions achieving recourse) for each method at time $\tau$. We repeat the procedure 10 times. In these experiments, we assume to know the true causal graph and structural equations.

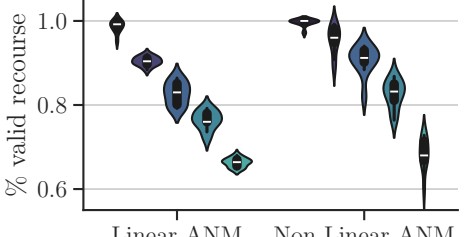

**Uncertainty invalidates counterfactual recourse (CAR) over time.** First, we consider a *stationary* version of the synthetic ANMs with exogenous noise $U_i \sim \mathcal{N}(0, \sigma_U)$ for all $i \in [3]$ and vary the variance of the exogenous factors $\sigma_U \in \{0, 0.3, 0.5, 0.7, 1.0\}$, where $\sigma_U = 0$ means that the ANM is deterministic. For each value of $\sigma_U$, we compute the recourse suggestions using robust counterfactual recourse (CAR). Fig. 3 displays how valid-

Figure 3: **Effect of uncertainty on counterfactual AR.** Empirical average validity and standard deviation over 10 runs of *robust* counterfactual algorithmic recourse (CAR) at $t = 50$. We vary the variance $\sigma_U$ of the exogenous factors of the stochastic process. Legend ($\sigma_U$): ■ 0 ■ 0.3 ■ 0.5 ■ 0.7 ■ 1.0.

ity decreases as variance increases showing that the validity over time of (robust) CAR recommendations is strongly impacted by the exogenous noise, as per Proposition 1, even in an ideal case in which the SCM is stationary and known. Appendix D shows extended results for $t \in \{0, 100\}$.

**Incorporating time is beneficial in causal algorithmic recourse.** Fig. 4 shows how T-SAR (Algorithm 1) achieves superior validity over time than robust (non-)causal methods on the synthetic

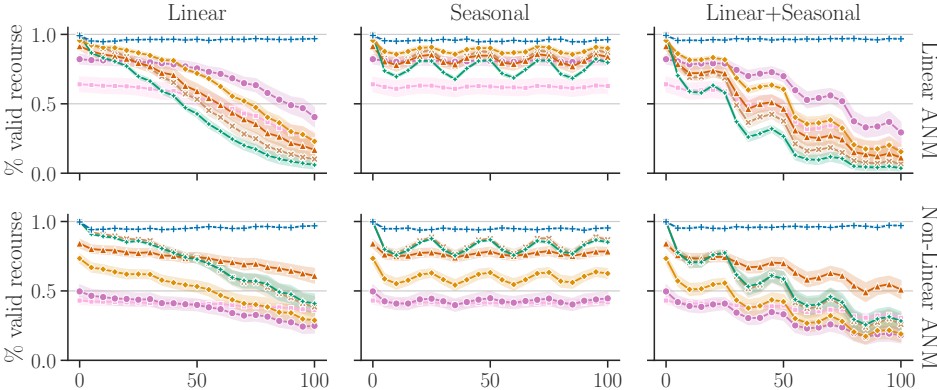

Figure 4: **Causal algorithmic recourse on diverse time series.** Empirical average validity and standard error (10 runs) for the robust ($\epsilon \in \{3, 5\}$) and time-aware causal recourse methods for the synthetic ANMs under different trends ($\alpha = 1.0$). Legend: ▬ T-SAR ▬ CAR ($\epsilon = 3$) ▬ SAR ($\epsilon = 3$) ▬ IMF ($\epsilon = 3$) and ▬ CAR ($\epsilon = 5$) ▬ SAR ($\epsilon = 5$) ▬ IMF ($\epsilon = 5$).

settings considering the diverse type of trends $m(t) \in \{\text{Linear}, \text{Seasonal}, \text{Linear+Seasonal}\}$. Interestingly, robust causal recourse methods depend highly on the chosen hyperparameters ($\epsilon$, $\eta$ and $\lambda$) while T-SAR requires less tuning. For example, both CAR and IMF show worse validity on the non-linear ANM when increasing $\epsilon$. Lastly, in Appendices E and F we provide further experiments on the *tradeoff* between *validity over time* and *cost*, and on the *sparsity* of recourses, respectively.

## 4.2 EXPERIMENT WITH REALISTIC TIME-SERIES

In Section 4.1, we assume perfect knowledge of the causal graph and structural equations governing the stochastic process. We now relax these assumptions by learning the structural equations in a data-driven manner, using a simple generative model, on three datasets.

**Experiments setup**. We consider three real-world datasets concerning high-risk decision tasks: recidivism prediction (COMPAS (Angwin et al., 2016)), and loan approval (Adult (Dua & Graff, 2017) and Loan (Karimi et al., 2021)). They involve categorical and continuous features, some of which are not actionable (*e.g.*, age, ethnicity, etc.). We use the causal graphs defined by Nabi & Shpitser (2018) for COMPAS and Adult, and Karimi et al. (2021) for Loan. We extend these datasets by adding linear+seasonal trends reflecting real-world phenomena (*e.g.*, income can fluctuate depending on the job market or individual expenses). Full details are available in Appendix C. Given the ground truth SCMs, we sample an additional separate time series with 2000 individuals for $t \in [0, 100]$, and we use all the samples up to $t = 50$ to learn an approximate SCM for each real-world dataset. We approximate the structural equations using a CVAE-like generative model (Sohn et al., 2015). In the experiment, all methods use the same approximate SCMs to compute recourse. Lastly, we perform the same evaluation procedure used for the synthetic experiments.

**Incorporating time is beneficial also with approximate SCMs.** Fig. 5 (top) shows how T-SAR provides better recourse recommendations than the robust counterparts by exploiting an estimator $\tilde{P}(\mathbf{X}^t)$. In both Adult and Loan, T-SAR shows equal or better validity than the non-temporal methods. For example, in Loan, T-SAR achieve almost twice the average validity ($\sim 72\%$) of the best non-temporal approach SAR ($\sim 39\%$) for $t = 50$. Understandably, T-SAR's performance hinges on the quality of the underlying estimator. In COMPAS, while T-SAR has good performance for $t \in [0, 20]$, it gracefully degrades afterwards. This occurs because the trend estimator it relies on underestimates the trend impact on the features. In fact, Appendix G shows that, if we employ a perfect estimator $P(\mathbf{X}^t)$, T-SAR outperforms all non-temporal methods (Fig. 12). As in Section 4.1, the non-temporal methods are sensitive to the hyperparameters, *e.g.*, for COMPAS, CAR provides higher validity for $\epsilon = 0.05$ rather than $\epsilon = 0.5$, while T-SAR needs less tuning.

**Accounting for time ensures more targeted interventions.** Lastly, Fig. 5 (bottom) shows how T-SAR suggests interventions counteracting the effect of the trend more effectively than non-temporal methods. We excluded COMPAS from the analysis since it has a single actionable feature. In Loan, we have two actionable features $\{income, savings\}$, with *income* subject to a trend.

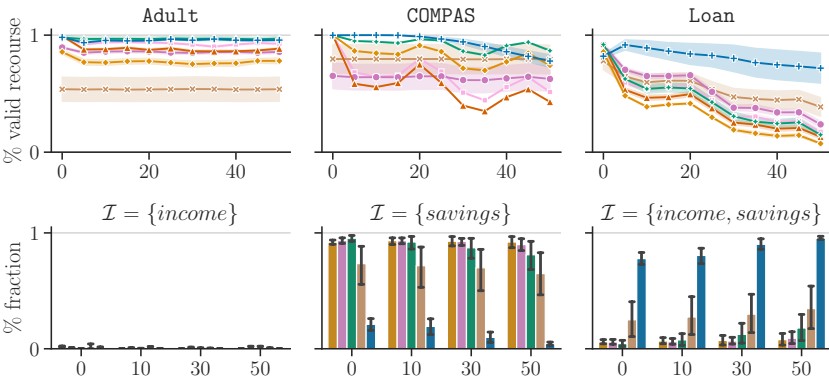

Figure 5: **Effect of time on realistic datasets.** (Top) Empirical average validity and standard error (10 runs) for the robust ($\epsilon \in \{0.05, 0.5\}$) and time-aware causal recourse methods for the realistic datasets under a non-linear trend. (Bottom) Distribution of the intervention sets $\mathcal{I}$ over the actionable features achieving recourse on `Loan` for different $t$. Legend: ■ T-SAR ■ CAR ($\epsilon = 0.05$) ■ SAR ($\epsilon = 0.05$) ■ IMF ($\epsilon = 0.05$) and ■ CAR ($\epsilon = 0.5$) ■ SAR ($\epsilon = 0.5$) ■ IMF ($\epsilon = 0.5$).

T-SAR provide recourse including the trend variable, while the other methods exclude it from the recommendation, thus yielding a lower validity. In `Adult`, we again have two actionable features {*education*, *work-hours-per-week*}, where *work-hours-per-week* is subject to a trend. In this case, T-SAR suggests acting on {*work-hours-per-week*} only. Robust sub-population methods (SAR) will instead ask the user to act on both {*education*, *work-hours-per-week*} because they have to robustify on both variables since they cannot forecast how they will change. Non-causal methods (IMF) act on all actionable features but achieve a lower validity since they cannot account for trend effects.

## 5 LIMITATIONS

We now discuss some limitations of our work which open up interesting avenues for future work.

**Feasibility of temporal recourse and its evaluation.** T-SAR depends on the quality of the estimator $\tilde{P}(\mathbf{X}^t)$ and, if the estimator is flawed, T-SAR could provide sub-optimal recourse. It is well-known how reliable time series forecasting is hard in various settings (Makridakis et al., 2020), because of issues like concept drift (Gama et al., 2014). Thus, it represents an additional hurdle to achieving practical temporal recourse for realistic applications. Our experiments on synthetic and semi-synthetic datasets are sufficient to confirm that time presents a non-trivial challenge for AR and to show how an estimator approximating $P(\mathbf{X}^t)$ can still be useful in some settings. However, we could not fully evaluate the effectiveness of T-SAR in real-world situations as this requires temporal datasets for recourse, which are currently not available. The scarcity of suitable data is a well-known issue affecting the evaluation of AR approaches at large (Karimi et al., 2021; Esfahani et al., 2024).

**Causal models, trends and interventions.** Our formalization assumes the stochastic process conforms to TiMiNo (Peters et al., 2013), leaving space to consider more complex SCMs. We do not explore trend models for the classifier $h$ or the cost function $C(\cdot)$, which could be present alongside those for $P(\mathbf{X}^t)$. Lastly, we assume the total causal effect of recourse can be observed within $t + \tau$, and future works could consider modelling interventions with causal effects extending beyond $t + \tau$.

## 6 CONCLUSION

We have investigated the impact of time on algorithmic recourse. Our formalization of temporal causal recourse extends both counterfactual and sub-population causal AR by modelling the world as a (possibly non-stationary) causal stochastic process $P(\mathbf{X}^t, Y^t)$. It allows us to theoretically demonstrate how *standard and robust AR approaches are fragile*, as their solutions become invalid in the presence of trends and future uncertainty. We also show that a simple algorithm, leveraging an estimator of the stochastic process fitted on historical data, can deliver more robust solutions. Our experiments with causal and non-causal approaches support our findings. With this work, we aim to highlight the negative impact of time on existing AR approaches while demonstrating how these challenges can be at least partially mitigated by leveraging historical data.

ETHICS STATEMENT

Our work aims at achieving *algorithmic contestability* (Lyons et al., 2021) via actionable counterfactual explanations, enabling users to overturn decisions taken by machine learning models. However, recourses present diverse challenges from both technical and ethical standpoints (Venkatasubramanian & Alfano, 2020). For example, fairness considerations (von Kügelgen et al., 2022) may arise from the applications of recourse methods, and they should be taken into consideration *before* considering real applications.

REPRODUCIBILITY STATEMENT

We report all the assumptions and proofs of the theorems, corollaries and propositions in Appendices B and H, making appropriate references to the relevant sections of the main paper (*e.g.*, Sections 3.2 to 3.5). In Appendix C we describe in-depth details regarding the experimental settings, the synthetic and realistic stochastic processes, the recourse methods, the generative model to learn approximate SCMs and the training pipeline adopted. We will release the source code and the raw results under a permissive license on `GitHub`. Currently, the code is available as an anonymized `.zip` in the supplementary material.

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

## A  Summary of the Theoretical Results on Temporal Algorithmic Recourse

Table 1: **Overview of the theoretical results and (non-)causal recourse methods.** We summarize the characteristics of the (non)-causal algorithmic recourse methods we considered in this work and the related theoretical results (Section 3). T-SAR is the only approach robust to time (upon possessing an estimator of the stochastic process). IMF is a non-causal method, so its recourse considers the *independently manipulable features* assumption *e.g.*, $P^{do(\mathbf{X}_\mathcal{I}=\mathbf{x}_\mathcal{I}+\boldsymbol{\theta})}(\mathbf{X}) = \mathbf{x}_\mathcal{I} + \boldsymbol{\theta}$.

| Method | Causality | Recourse | Robust to time |
|---|---|---|---|
| IMF (Wachter et al., 2017) | - | Individualized | ✗ (by Corollary 4 and Proposition 5) |
| CAR (Karimi et al., 2020b) | Counterfactual | Individualized | ✗ (by Corollary 4 and Proposition 5) |
| SAR (Karimi et al., 2020b) | Interventional | Sub-population | ✗ (by Corollary 4 and Proposition 5) |
| T-CAR (Definition 1) | Counterfactual | Individualized | ✗ (by Proposition 1 and Corollary 2) |
| T-SAR (Definition 1) | Interventional | Sub-population | ✓ (upon having an estimator $\tilde{P}(\mathbf{X}^t)$) |

## B  Proofs

### B.1  Proof of Proposition 1

*Proof.* Consider a TiMINo stochastic process $\{\mathbf{X}_t\}_{t\in\mathbb{N}}$ with structural equations:

$$X_i^t = f(\mathbf{Pa}_i^{t-p}, \ldots, \mathbf{Pa}_i^t) + U_i^t, \quad U_i^t \sim \mathcal{N}(\mu_X, \sigma_X), \quad \mu_X, \sigma_X > 0 \tag{10}$$

where $U_i^t$ have positive mean and variance. We prove the proposition by contradiction by looking at the value of the time lag $\tau$. For the sake of clarity, we state the proof for $p = 1$ (the proof for $p > 1$ is similar). Assume that we observed a sequence of realizations $\mathbf{x}^{t-p:t}$ and we can compute the counterfactual distribution $P^{do(\mathbf{X}^{t+\tau}=\mathbf{x}^{t+\tau}+\boldsymbol{\theta}),\mathbf{X}^{t-p:t}=\mathbf{x}^{t-p:t}}(\mathbf{X}^{t+\tau})$ for any $\tau \geq 0$ and $\boldsymbol{\theta} \in \mathbb{R}^d$. If $\tau = 0$, we can immediately recover the value of all exogenous factors via $u_i^t = x_i^t - f(\mathbf{Pa}_i^{t-p}, \ldots, \mathbf{Pa}_i^t)$, as we know $\mathbf{x}^{t-p:t}$. Let us assume we can also recover the exogenous factors for a time lag $\tau > 0$ even if we did not observe future realizations $\{\mathbf{x}^t, \ldots, \mathbf{x}^{t+\tau}\}$. Since we did not observe such realizations, the classical abduction step to recover the exogenous factors within the $[t, t+\tau]$ interval is impossible. Recently, Bynum et al. (2023) introduced *forward-looking counterfactuals* to overcome this challenge. They postulate we can still compute counterfactuals over unseen realizations by *propagating* the latest exogenous factors we were able to abduce (Bynum et al., 2023, Section 3). Similary, we assume we can *propagate* the latest exogenous factors we can recover ($\mathbf{u}^t$) into the future, *e.g.*, $U_i^{t+\tau} = u_i^t$ for any $\tau > 0$ and $i \in [d]$. Since we assume we can compute the exact counterfactual distribution, the only setting where $U_i^{t+\tau} = u_i^t$ holds is if $\sigma_X = 0$ and $\mu_X = 0$. However, we initially stated that $\mu_X, \sigma_X > 0$, so we have a contradiction. □

### B.2  Proof of Proposition 3

*Proof.* Consider a discrete-time stationary time series $\{(\mathbf{x}, y)^t\}_{t\in\mathbb{N}}$ and wlog consider a fixed linear classifier $h(\mathbf{x}) = \boldsymbol{\beta}^\top \mathbf{x}$ that is *injective*, that is, $\mathbf{x} \neq \mathbf{x}' \Rightarrow h(\mathbf{x}) \neq h(\mathbf{x}')$. Given a realization $\mathbf{x}^t$, let us assume $\boldsymbol{\theta}^*$ is the optimal intervention for Eq. (2) at time $t$, but not at time $t + \tau$, with $\tau \in \mathbb{N}$. Thus, either our recourse becomes invalid or exceedingly expensive. We focus on the former since we assume a fixed cost function such as the $\ell_1$-norm of the intervention $C(\boldsymbol{\theta}) = \|\boldsymbol{\theta}\|$. As a consequence, we have $\mathbb{E}[h(\hat{\mathbf{x}}^{t+\tau}) - h(\hat{\mathbf{x}}^t)] \neq 0$ where $\hat{\mathbf{x}}^{t+\tau} \sim P^{do(\boldsymbol{\theta}^*)}(\mathbf{X}^{t+\tau} \mid \mathbf{X}^{t+\tau}_{nd(\mathcal{I})} = \mathbf{x}^{t+\tau}_{nd(\mathcal{I})})$ and $\hat{\mathbf{x}}^t \sim P^{do(\boldsymbol{\theta}^*)}(\mathbf{X}^t \mid \mathbf{X}^t_{nd(\mathcal{I})} = \mathbf{x}^t_{nd(\mathcal{I})})$. We denote with $\mathcal{I} \subseteq [d]$ the intervention set of the successful intervention $\boldsymbol{\theta}^*$, and with $do(\boldsymbol{\theta}^*) = do(\mathbf{X}_\mathcal{I} = \mathbf{x}_\mathcal{I} + \boldsymbol{\theta}^*)$ the corresponding soft intervention. Consider the case in which our intervention does not achieve recourse after $\tau$ time steps. We can decompose the previous expectation as follows:

$$\mathbb{E}[h(\hat{\mathbf{x}}^{t+\tau}) - h(\hat{\mathbf{x}}^t)] = \mathbb{E}[\boldsymbol{\beta}^\top \hat{\mathbf{x}}^{t+\tau} - \boldsymbol{\beta}^\top \hat{\mathbf{x}}^t] \tag{11}$$

$$= \boldsymbol{\beta}^\top \mathbb{E}[\hat{\mathbf{x}}^{t+\tau} - \hat{\mathbf{x}}^t] \tag{12}$$

Thus, since $\boldsymbol{\theta}^*$ is not optimal for $t + \tau$, we must have $\mathbb{E}[\hat{\mathbf{x}}^{t+\tau} - \hat{\mathbf{x}}^t] \neq 0$. This, however, is a contradiction since we assume the time series is stationary. It follows that, since $\tau$ is arbitrary, for any $t$, the corresponding optimal solution $\boldsymbol{\theta}^*$ is also optimal for all $\tau > 0$. Now, this also means that $\boldsymbol{\theta}^*$ is also an optimal solution for the classical recourse optimization problem Eq. (1) as long as we optimize Eq. (1) by considering $P(\mathbf{X}, Y) = P(\mathbf{X}^t, Y^t)$ for any time steps $t \in \mathbb{N}$. $\qquad\square$

### B.3 FULL DERIVATIONS FOR EXAMPLE 1

*Proof.* Consider a trend-stationary stochastic process defined by these structural equations:

$$
\begin{aligned}
X^t &= \alpha X^{t-1} + m(t) + U_X^t, & U_X^t &\sim \mathcal{N}(\mu_X, \sigma_X) \\
Y^t &= \beta X^t + U_Y^t, & U_Y^t &\sim \mathcal{N}(0, 1)
\end{aligned}
\tag{13}
$$

for all $t$, $\alpha \in (0, 1)$ and $\beta \in \mathbb{R}$. The function $m(t) : \mathbb{R} \to \mathbb{R}$ represents a *trend* independent of $X^t$ and $Y^t$. We consider a linear trend $m(t) = -ct + U_m^t$, where $U_m^t \sim \mathcal{N}(\mu_m, \sigma_m)$ and $c \in \mathbb{R}^+$. Given a realization $x^{t-1}$, the state of $X^{t+\tau}$ admits the closed-form expression:

$$
X^t = \alpha x^{t-1} + m(t) + U_X^t
\tag{14}
$$

$$
X^{t+1} = \alpha^2 x^{t-1} + \alpha m(t) + \alpha U_X^t + m(t+1) + U_X^{t+1}
\tag{15}
$$

$$
X^{t+2} = \alpha^3 x^{t-1} + \alpha^2 m(t) + \alpha^2 U_X^t + \alpha m(t+1) + \alpha U_X^{t+1} + m(t+2) + U_X^{t+2}
\tag{16}
$$

$$
\vdots
\tag{17}
$$

$$
X^{t+\tau} = \alpha^{\tau+1} x^{t-1} + \sum_{i=0}^{\tau} \alpha^{\tau-i} \left( m(t+i) + U_X^{t+i} \right)
\tag{18}
$$

Hence, the expectation of $Y^{t+\tau}$ with respect to the interventional distribution $P^{do(\theta)}(X^t, Y^t)$ is:

$$
\begin{aligned}
\mathbb{E}[Y^{t+\tau}] &= \mathbb{E}[\beta X^{t+\tau} + U_Y^{t+\tau}] \\
&= \beta \mathbb{E}[X^{t+\tau}] + \mathbb{E}[U_Y^{t+\tau}] \\
&\overset{(i)}{=} \beta \mathbb{E}[\alpha^{\tau+1} x^{t-1} + \textstyle\sum_{i=0}^{\tau} \alpha^{\tau-i} \left( m(t+i) + U_X^{t+i} \right)] \\
&\overset{(i)}{=} \beta \alpha^{\tau+1} x^{t-1} + \textstyle\sum_{i=0}^{\tau} \alpha^{\tau-i} \left( \mathbb{E}[m(t+i)] + \mathbb{E}[U_X^{t+i}] \right) \\
&= \beta \left( \alpha^{\tau+1} x^{t-1} + \textstyle\sum_{i=0}^{\tau} \alpha^{\tau-i} (-c(t+i) + \mu_m + \mu_X) \right)
\end{aligned}
$$

Here, $(i)$ follows by construction since $\mathbb{E}[U_Y^{t+\tau}] = 0$ for all $t$, $\tau$ and $i \in [d]$. We now consider the following fixed classifier $\sigma(Y^t \mid X^t)$ where $\sigma(x) = 1/(1 + e^{-x})$. Thus, the expectation over the classifier output becomes:

$$
\mathbb{E}[h(X^{t+\tau})] = \sigma \left( \beta \left( \alpha^{\tau+1} x^{t-1} + \textstyle\sum_{i=0}^{\tau} \alpha^{\tau-i} (-c(t+i) + \mu_m + \mu_X) \right) \right)
\tag{19}
$$

Given that we are considering soft interventions, we consider the cost function $C(\hat{x}^{t+\tau}, x^{t+\tau}) = \hat{x}^{t+\tau} - x^{t+\tau} = \theta$ since $\hat{x}^{t+\tau} = x^{t+\tau} + \theta$. Given that $\sigma(x) \geq 1/2$ if and only if $x \geq 0$, we have that the optimal intervention $\theta^{t+\tau} \in \mathbb{R}$ for which we have $\mathbb{E}[h(X^{t+\tau} + \theta)] \geq 1/2$ can be expressed as:

$$
\theta^{t+\tau} = -\alpha^{\tau+1} x^{t-1} - \textstyle\sum_{i=0}^{\tau} \alpha^{\tau-i} (-c(t+i) + \mu_m + \mu_X)
\tag{20}
$$

$\qquad\square$

### B.4 PROOF OF PROPOSITION 5

We can prove Proposition 5 by showing how we can *always* find a simple trend invalidating *any* (robust) intervention.

*Proof.* Let us consider a trend-stationary stochastic process $P(\mathbf{X}^t, Y^t)$ and fixed injective classifier $h$ approximating $P(\mathbf{X}^t \mid Y^t)$. We denote with $\mathbf{m}(t) : \mathbb{N}^d \to \mathbb{R}^d$ the $d$-variate trend function where $m_i(t)$ is the trend component for a single random variable $X_i^t$ for any $i \in [d]$ and $t \in \mathbb{N}$. Given a negatively classified instance $\mathbf{x}^t$, assume $\boldsymbol{\theta}$ is the optimal robust intervention for a fixed $\epsilon > 0$ and for the timestep $t$.

Consider the following trend function $\mathbf{m}(t) = \mathbf{1}\{t \geq \tau\}(-\boldsymbol{\theta})$ which is adding the inverse of the optimal intervention if an only if $t \geq \tau$. Specifically, we define each trend component as $m_i(t) = \mathbf{1}\{t \geq \tau\}(-\theta_i)$. If our stochastic process exhibits such a trend, then, for any fixed $\tau > 0$, the robust intervention is invalid *e.g.*, $\mathbb{E}[h(\hat{\mathbf{x}}^{t+\tau})] < 1/2$. Moreover, we can always build such a trend for any $\boldsymbol{\theta}$ and any trend-stationary stochastic process.

$\square$

## B.5 PROOF OF THEOREM 6

*Proof.* We first apply the following substitutions (a) $\boldsymbol{\beta}' = \boldsymbol{\beta}^{t+\tau}$ and $\mathbf{x}' = \hat{\mathbf{x}}^{t+\tau}$ (b) $\boldsymbol{\beta} = \boldsymbol{\beta}^t$ and $\mathbf{x} = \hat{\mathbf{x}}^t$, to improve the clarity of the proof. Then, the proof is the following:

$$
\begin{aligned}
\mathbb{E}[|h^{t+\tau}(\hat{\mathbf{x}}^{t+\tau}) - h^t(\hat{\mathbf{x}}^t)|] &= \mathbb{E}[|\langle\boldsymbol{\beta}', \mathbf{x}'\rangle - \langle\boldsymbol{\beta}, \mathbf{x}\rangle|] \\
&= \mathbb{E}[|\langle\boldsymbol{\beta}', \mathbf{x}'\rangle + \langle\boldsymbol{\beta}, \mathbf{x}'\rangle - \langle\boldsymbol{\beta}, \mathbf{x}'\rangle - \langle\boldsymbol{\beta}, \mathbf{x}\rangle + \langle\boldsymbol{\beta}, \mathbf{x}'\rangle - \langle\boldsymbol{\beta}, \mathbf{x}'\rangle|] \\
&= \mathbb{E}[|\langle\boldsymbol{\beta}' - \boldsymbol{\beta}, \mathbf{x}'\rangle + \langle\boldsymbol{\beta}, \mathbf{x}'\rangle + \langle\boldsymbol{\beta}, \mathbf{x}' - \mathbf{x}\rangle - \langle\boldsymbol{\beta}, \mathbf{x}'\rangle|] \\
&\leq \mathbb{E}[|\langle\boldsymbol{\beta}' - \boldsymbol{\beta}, \mathbf{x}'\rangle|] + \mathbb{E}[|\langle\boldsymbol{\beta}, \mathbf{x}' - \mathbf{x}\rangle|] \\
&\overset{(i)}{\leq} \mathbb{E}[\|\boldsymbol{\beta}' - \boldsymbol{\beta}\| \cdot \|\mathbf{x}'\|] + \mathbb{E}[\|\boldsymbol{\beta}\| \cdot \|\mathbf{x}' - \mathbf{x}\|] \\
&\overset{(ii)}{\leq} k\sqrt{d} \cdot \mathbb{E}[\|\boldsymbol{\beta}' - \boldsymbol{\beta}\|] + k\sqrt{d} \cdot \mathbb{E}[\|\mathbf{x}' - \mathbf{x}\|] \\
&= k\sqrt{d} \cdot \left(\mathbb{E}\left[\|\boldsymbol{\beta}^{t+\tau} - \boldsymbol{\beta}^t\|\right] + \mathbb{E}\left[\|\hat{\mathbf{x}}^{t+\tau} - \hat{\mathbf{x}}^t\|\right]\right)
\end{aligned}
$$

where (*i*) follows from the Cauchy-Schwarz inequality and (*ii*) from the bounds we placed on $\mathbf{X}$ and $\boldsymbol{\beta}$ (*e.g.*, since $-\mathbf{k} \leq \boldsymbol{\beta} \leq \mathbf{k}$ and since $|\boldsymbol{\beta}| = d$ we have $\max_{\boldsymbol{\beta}}\|\boldsymbol{\beta}\| = k\sqrt{d}$). Moreover, since $h^t$ is trained over a fixed dataset $\mathcal{D}^t$, we have that $\boldsymbol{\beta}^t \perp \mathbf{X}^t \mid \mathcal{D}^t$ for all $t \in \mathbb{N}$. We stress that the bounds placed on $\mathbf{X}$ and $\boldsymbol{\beta}$ enable us to constrain the $\Delta h(\boldsymbol{\theta}; \tau)$ variation. In the unbounded case, where $k \to \infty$, clearly no upper bound is possible. $\square$

## B.6 PROOF OF COROLLARY 7

We begin the proof of Corollary 7 by starting from the previous proof for Theorem 6 (given in Appendix B.5). Please recall that a stochastic process $P(\mathbf{X}^t)$ is trend-stationary when it can be expressed as $\mathbf{X}^t = \mathbf{m}(t) + \mathbf{e}^t$, where $\mathbf{m}(t)$ is a (non-)linear trend function and $\mathbf{e}^t$ is a stationary stochastic process. In the following, we denote with $\tilde{\mathbf{x}}$ the stationary part of the stochastic process, and we consider *deterministic* trend functions.

*Proof.* Let us assume that each random variable $X_i^t$ can be described as a trend-stationary univariate stochastic process. Thus, let us define with $\mathbf{m}(t) = \{m_i(t)\}_{i=1}^d$ the trend function, where $m_i(t)$ the trend component for the $i$-th variable. Then, we define as $m^*(t) = \max_{i\in[d]} m_i(t)$ the largest trend for $t \in \mathbb{N}$. The derivation for the upper bound is the following:

$$
\begin{aligned}
\mathbb{E}[|h^{t+\tau}(\hat{\mathbf{x}}^{t+\tau}) - h^t(\hat{\mathbf{x}}^t)|] &\leq k\sqrt{d} \cdot \left(\mathbb{E}\left[\|\boldsymbol{\beta}^{t+\tau} - \boldsymbol{\beta}^t\|\right] + \mathbb{E}\left[\|\hat{\mathbf{x}}^{t+\tau} - \hat{\mathbf{x}}^t\|\right]\right) \quad \text{(Theorem 6)} \\
&\overset{(i)}{=} k\sqrt{d} \cdot \left(\mathbb{E}\left[\|\boldsymbol{\beta}^{t+\tau} - \boldsymbol{\beta}^t\|\right] + \mathbb{E}\left[\|\tilde{\mathbf{x}}^{t+\tau} + \mathbf{m}(t+\tau) - \tilde{\mathbf{x}}^t - \mathbf{m}(t)\|\right]\right) \\
&= k\sqrt{d} \cdot \left(\mathbb{E}\left[\|\boldsymbol{\beta}^{t+\tau} - \boldsymbol{\beta}^t\|\right] + \mathbb{E}\left[\|\tilde{\mathbf{x}}^{t+\tau} - \tilde{\mathbf{x}}^t\|\right] + \mathbb{E}\left[\|\mathbf{m}(t+\tau) - \mathbf{m}(t)\|\right]\right) \\
&\overset{(ii)}{=} k\sqrt{d} \cdot \left(\mathbb{E}\left[\|\boldsymbol{\beta}^{t+\tau} - \boldsymbol{\beta}^t\|\right] + \mathbb{E}\left[\|\mathbf{m}(t+\tau) - \mathbf{m}(t)\|\right]\right) \\
&\overset{(iii)}{\leq} k\sqrt{d} \cdot \left(\mathbb{E}\left[\|\boldsymbol{\beta}^{t+\tau} - \boldsymbol{\beta}^t\|\right] + \sqrt{d}(m^*(t+\tau) - m^*(t))\right) \\
&= k\sqrt{d} \cdot \mathbb{E}\left[\|\boldsymbol{\beta}^{t+\tau} - \boldsymbol{\beta}^t\|\right] + kd\left(m^*(t+\tau) - m^*(t)\right)
\end{aligned}
$$

$$\tag{21}$$

where (*i*) and (*ii*) follows from the definition of a *trend-stationary* stochastic process. Then, (*iii*) follows since we can substitute each univariate trend $m_i(t)$ with the maximum $m^*(t)$ for the time step $t$ within the $\ell_2$-norm. $\square$

## C IMPLEMENTATION DETAILS FOR THE EXPERIMENTS WITH SYNTHETIC AND REAL DATA

In this section, we describe the synthetic and realistic stochastic processes we used in our experiments (cf. Section 4). We also describe the training steps and generative model used to approximate the SCMs. Lastly, we report several technical implementation details.

### C.1 SYNTHETIC ADDITIVE TREND FUNCTION.

We want our causal stochastic process to describe an environment where the changes to obtain a positive classification fluctuate over time. For example, as an individual ages, it will become less likely for her to repay her loan in the case of a loan application (based on the survival rate of the population). We define an additive trend function $m(t) : \mathbb{N} \to \mathbb{R}^+$, which is a linear combination between a *linear* and *seasonal* trend. We control the mixture of each component with two parameters, $\beta_l \in \mathbb{R}^+$ and $\beta_s \in \mathbb{R}^+$, respectively. We also consider an additional parameter $\alpha \in [0, 1]$ controlling the trend's strength over the stationary causal process.

$$m(t) = \alpha \cdot (\beta_l \cdot \min(0.05 \cdot t, 10) + \beta_s \cdot |\sin(0.5 \cdot t)|) \tag{22}$$

In our experiments, we set $\beta_l \in \{0, 1\}$ and $\beta_s \in \{0, 1.5\}$ for the linear ANM, and $\beta_l \in \{0, 2\}$ and $\beta_s \in \{0, 5\}$ for the non-linear ANM. For the realistic experiments, we set $\beta_l, \beta_s \in \{0, 1\}$ for Adult, $\beta_l \in \{0, 0.3\}$ and $\beta_s \in \{0, 1\}$ for COMPAS, and $\beta_l \in \{0, 0.5\}$ and $\beta_s \in \{0, 5\}$ for Loan.

### C.2 CAUSAL GRAPHS FOR THE EXPERIMENTS

For the synthetic experiments, we considered the synthetic 3-variables additive noise models (ANMs) from Karimi et al. (2020b). We extended them by transforming them into trend-stationary stochastic processes, by adding an autoregressive component and the trend $m(t)$ on the 3rd feature. If $\alpha = 0$, both ANMs give rise to stationary time series.

**Linear Additive Noise Model.**

$$
\begin{aligned}
X_1^t &= 0.5 \cdot X_1^{t-1} + U_1^t & U_1^t &\sim \text{MoG}(\mathcal{N}(-1, 1.5), \mathcal{N}(1, 1)) \\
X_2^t &= 0.5 \cdot X_2^{t-1} - 0.25 \cdot X_1^t + U_2^t & U_2^t &\sim \mathcal{N}(0, 0.1) \\
X_3^t &= 0.5 \cdot X_3^{t-1} + 0.05 \cdot X_1^t + 0.25 \cdot X_1^t - m(t) + U_3^t & U_3^t &\sim \mathcal{N}(0, 1)
\end{aligned}
\tag{23}
$$

**Non-linear Additive Noise Model.**

$$
\begin{aligned}
X_1^t &= 0.5 \cdot X_1^{t-1} + U_1^t & U_1^t &\sim \text{MoG}(\mathcal{N}(-2, 1.5), \mathcal{N}(1, 1)) \\
X_2^t &= 0.5 \cdot X_2^{t-1} - 1\frac{3}{1 + e^{-2X_1^t}} + U_2^t & U_2^t &\sim \mathcal{N}(0, 0.1) \\
X_3^t &= 0.5 \cdot X_3^{t-1} + 0.05 \cdot X_1^t + 0.25 \cdot (X_1^t)^2 - m(t) + U_3^t & U_3^t &\sim \mathcal{N}(0, 1)
\end{aligned}
\tag{24}
$$

**Label function.** We also consider the following conditional distribution $P(Y^t \mid \mathbf{X}^t)$, again taken from Karimi et al. (2020b), which produces roughly balanced groups:

$$Y^t \sim \text{Binomial}\left(1/\left(1 + \exp(-2.5 \cdot (X_1^t + X_2^t + X_3^t)/\rho)\right)\right) \tag{25}$$

where $\rho$ is the empirical mean of $X_1^t + X_2^t + X_3^t$ for $t = 0$. We use the label function only to train the classifier $h$ at time $t = 0$, then it is discarded and we rely only on $h$ for our experiments.

### C.3 CAUSAL GRAPHS FOR THE REALISTIC EXPERIMENTS

We now describe the design choices for the realistic datasets. We tried to find a balance between realism and simplicity, to provide scenarios close to potential real-world situations, that, however, we can easily control for our experiments. Therefore, some of the design choices might not represent faithfully how the system can evolve in real life.

**Adult (Dua & Graff, 2017).** We use the features and causal graph defined by Nabi & Shpitser (2018). Eq. (26) shows the full causal graph. We have the following features: $S$ (sex), $A$ (age),

$US$ (resident of the United States of America), $M$ (married), $E$ (education level) and $H$ (working hours per week). The features $S$, $US$, and $M$ are categorical variables, the others are represented as continuous random variables. We assume $S$, $A$, $US$ and $M$ remain fixed over time. The only actionable features are the education level $E$, and working hours per week $H$. We apply a decreasing trend to $H$. We employ non-linear structural equations $f_M$, $f_E$ and $f_H$ by using pre-trained 3-layer MLPs, trained on the original dataset, from Dominguez-Olmedo et al. (2022). Moreover, as a label function, we also employ a classifier $h$ taken by Dominguez-Olmedo et al. (2022) to label the examples at $t = 0$.

$$
\begin{aligned}
S^t &= \mathbf{1}\{t > 0\} \cdot S^{t-1} + \mathbf{1}\{t = 0\} \cdot U_S & U_S &\sim \text{Binomial}(0.9) \\
A^t &= \mathbf{1}\{t > 0\} \cdot U^{t-1} + \mathbf{1}\{t = 0\} \cdot U_A^t & U_A^t &\sim \mathcal{N}(0, 1) \\
US^t &= \mathbf{1}\{t > 0\} \cdot US^{t-1} + \mathbf{1}\{t = 0\} \cdot U_{US} & U_{US} &\sim \text{Binomial}(0.9) \\
M^t &= \mathbf{1}\{\sigma(f_M(S^t, A^t, US^t)) > 1/2\} \\
E^t &= 0.5 \cdot E^{t-1} + f_E(S^t, A^t, US^t, M^t) + U_E & U_E &\sim \mathcal{N}(0, 1) \\
H^t &= 0.5 \cdot H^{t-1} + f_H(S^t, A^t, US^t, M^t) - m(t) + U_H & U_H &\sim \mathcal{N}(0, 1)
\end{aligned}
\tag{26}
$$

**COMPAS (Angwin et al., 2016).** We use the features and causal graph defined by Nabi & Shpitser (2018). Eq. (27) shows the full causal graph. We have the following features: $S$ (sex), $A$ (age), $C$ (ethnicity, if caucasian or not), and $P$ (prior counts). The feature $S$ is categorical, and the others are represented as continuous random variables. We assume $S$, $A$, and $C$ remain fixed over time, and that the only actionable feature is the prior count $P$. We apply an increasing trend to $P$. As we did for Adult, we obtain the pre-trained non-linear structural equations $f_C$ and $f_P$ and label function from Dominguez-Olmedo et al. (2022).

$$
\begin{aligned}
S^t &= \mathbf{1}\{t > 0\} \cdot S^{t-1} + \mathbf{1}\{t = 0\} \cdot U_S & U_S &\sim \text{Binomial}(0.8) \\
A^t &= \mathbf{1}\{t > 0\} \cdot U^{t-1} + \mathbf{1}\{t = 0\} \cdot U_A^t & U_A^t &\sim \text{Poisson}(1) \\
C^t &= \mathbf{1}\{t > 0\} \cdot C^{t-1} + \mathbf{1}\{t = 0\} \cdot \left(\sigma(f_C(S^t, A^t)) + U_C^t\right) & U_C^t &\sim \mathcal{N}(0, 1) \\
P^t &= 0.5 \cdot P^{t-1} + f_P(S^t, A^t, C^t) + m(t) + U_P^t & U_P^t &\sim \mathcal{N}(0, 1)
\end{aligned}
\tag{27}
$$

**Loan (Karimi et al., 2020b).** We use the causal graph defined by Karimi et al. (2020b) presenting a semi-synthetic loan approval scenario inspired by the German Credit dataset (Hofmann, 1994). For the structural equations, we use the one adapted by Dominguez-Olmedo et al. (2022) in their paper. Eq. (28) shows the full causal graph. We have the following features: $G$ (gender), $A$ (age), $E$ (education level), $L$ (loan amount), $D$ (loan duration), $I$ (income) and $S$ (savings). $G$ is a categorical variable, while the rest are considered continuous. Moreover, $G$ remains fixed over time. We assume the only actionable features are $S$ and $I$. We apply a trend to the income $I$.

$$
\begin{aligned}
G^t &= \mathbf{1}\{t > 0\} \cdot G^{t-1} + \mathbf{1}\{t = 0\} \cdot U_G^t & U_G^t &\sim \text{Binomial}(0.5) \\
A^t &= 0.5 \cdot A^{t-1} - 35 + U_A^t & U_A^t &\sim \text{Gamma}(10, 3.5) \\
E^t &= 0.5 \cdot E^{t-1} - 0.5 + \left(1 + e^{-\left(-1 + 0.5G^t + \left(1 + e^{-0.1A^t}\right)^{-1} + U_E\right)}\right)^{-1} & U_E &\sim \mathcal{N}(0, \sqrt{0.25}) \\
L^t &= 0.5 \cdot L^{t-1} + 1 + 0.01(A^t - 5)(5 - A^t) + G^t + U_L, & U_L &\sim \mathcal{N}(0, 2) \\
D^t &= 0.5 \cdot D^{t-1} - 1 + 0.1A^t + 2G^t + L^t + U_D & U_D &\sim \mathcal{N}(0, 3) \\
I^t &= 0.5 \cdot I^{t-1} - 4 + 0.1(A^t + 35) + 2G^t + G^t E^t + U_I - m(t) & U_I &\sim \mathcal{N}(0, 2) \\
S^t &= 0.5 \cdot S^{t-1} - 4 + 1.5 \cdot \mathbf{1}\{I^t > 0\} \cdot I^t + U_S & U_S &\sim \mathcal{N}(0, 5)
\end{aligned}
\tag{28}
$$

We sample the labels from the function defined by Karimi et al. (2020b):

$$
Y^t \sim \text{Bernoulli}\left(\left(1 + e^{-0.3(-L^t - D^t + I^t + S^t + IS^t)}\right)^{-1}\right).
\tag{29}
$$

## C.4 ON LEARNING AN APPROXIMATE SCM

We now describe the simple generative model we used in the experiment in Section 4.2. Similarly to Karimi et al. (2020b) and Dominguez-Olmedo et al. (2022), we *approximate* the structural equations in a data-driven manner. We assume we can represent the actionable features $X_i$ as Gaussian random variables $\mathcal{N}(\mu_i^t, 1)$ with constant variance and time-dependent $\mu^t$. For each random variable, we define the mean as the output of a regressor $f_i$ taking as input: the autoregressive component $\mathbf{X}_i^{t-1}$, the parents $\mathbf{Pa}_i^t$ and the time $t$. Thus, we obtain the following structural equations:

$$X_i^t = \mu_i^t + U_i^t \qquad U_i^t \sim \mathcal{N}(0,1) \quad \mu_i^t = f_i(X_i^{t-1}, \mathbf{Pa}_i^t, t) \tag{30}$$

Similarly to a conditional VAE (Sohn et al., 2015), we can both sample new instances from the approximate SCMs, but we can also compute the interventional or counterfactual distributions.

## C.5 TECHNICAL DETAILS AND CODE

We based our implementation of CAR, SAR, IMF and T-SAR following *adversarial robust algorithmic recourse* (Dominguez-Olmedo et al., 2022). Namely, we leveraged their implementation[5] and we adapted their code to work with time-based uncertainty sets (cf. Section 3.6). Moreover, we also used their causal graph implementations, pre-trained models, and preprocessing steps as a starting point for building our stochastic processes. In the case of the synthetic datasets, we instead looked at Karimi et al. (2020b) original implementation[6]. Our code and experimental results will be released on Github with a permissive license[7]. Lastly, we run our experiments on a Linux machine (Ubuntu 22.04, 4 LTS) with 32 CPU cores and 125 GB of RAM. Our implementation is written in Python 3.10, using standard scientific and deep learning libraries such as numpy (Harris et al., 2020) and PyTorch (Paszke et al., 2019). The various hyperparameters are duly specified in the source code, but we report the most important here for clarity.

**Algorithm 1 hyperparameters.** We approximate $B(\mathbf{x}^t; \tau)$ by sampling only 20 instances for all datasets and we set the number of *epochs* to $N = 30$. As a penalty, we set $\lambda = 1$ for the Lagrangian (line 5, Algorithm 1). Please notice that Dominguez-Olmedo et al. (2022) uses a *decaying rate* to reduce the impact of the cost function on the loss $\mathcal{L}$ after each epoch (we kept the original hardcoded value of 0.02). As learning rate, we set $\eta = 0.5$ for the synthetic experiments, and $\eta = 3$ for the realistic datasets. The learning rate is the same for all the methods. We did not perform a full grid search over the parameter space, since we found empirically our chosen hyperparameters were giving satisfactory performances.

**Classifiers $h$.** For each setting, we trained a 3-layered MLP approximating $P(Y^t \mid \mathbf{x}^t)$, via *empirical risk minimization* by sampling a given dataset for $t = 0$. We use stochastic gradient descent (SGD) to minimize the binary cross entropy loss $\mathcal{L} = -\frac{1}{N} \sum_{\mathbf{x}^0, y^0 \in \text{batch}} (y^0 \log h(\mathbf{x}^0) + (1 - y^0) \log (1 - h(\mathbf{x}^0))$ (*e.g.*, torch.nn.BCELoss[8]) where $y^0$ is the ground truth label. In our experiments, we set the batch size to 100, the number of epochs to 15 and the learning rate to 0.001, for all datasets. The accuracy of the models for a single seed are: 0.847 (Linear ANM), 0.963 (Non-Linear ANM), 0.817 (Adult), 0.645 (COMPAS) and 0.842 (Loan).

**Approximate structural equations.** In our experiments, we consider only linear $f_i$. We train each $f_i$ via *empirical risk minimization* following the procedure outlined in Section 4.2. For each feature $i \in [d]$, and for each epoch, we consider a batch $\{(x_i^t, x_i^{t-1}, \mathbf{Pa}_i^t, t)_j\}_{j=1}^b$ and we minimize the *mean squared error* between the ground truth $x_i^t$ and the model output $f_i(X_i^{t-1}, \mathbf{Pa}_i^t, t)$ with stochastic gradient descent. We fix the batch size to 100, the learning rate to 0.001 and the number of epochs to 15 for all settings. We report here the *mean squared error* over 50 timesteps (2000 individuals) for the approximate SCMs we used in Section 4.2. We compute the MSE for each feature for each timestep, and then we average. The empirical average MSE and standard deviation over 10 runs is: $1.162 \pm 0.005$ (Adult), $258.226 \pm 7.328$ (COMPAS) and $10.447 \pm 0.037$ (Loan).

---

[5] https://github.com/RicardoDominguez/AdversariallyRobustRecourse
[6] https://github.com/charmlab/recourse
[7] https://github.com./xxxx/xxxxx. The code and experimental results are provided as a .zip file in the Supplementary Material as instructed by the ICLR guidelines.
[8] https://pytorch.org/docs/stable/generated/torch.nn.BCELoss.html

# D ADDITIONAL EMPIRICAL RESULTS ON THE EFFECT OF UNCERTAINTY ON COUNTERFACTUAL RECOURSE OVER TIME

We present the extended results of the experiment measuring empirically the impact of the uncertainty in CAR (Section 3.2). The experimental setting and evaluation procedure are the same as Section 4.1. Fig. 6 shows the empirical average validity of CAR's recourse over time $t \in \{0, 100\}$. The plot shows how uncertainty heavily impacts the recourse validity from the initial time steps as $\sigma_{\mathbf{U}}$ grows, even when $P(\mathbf{X}^t)$ is stationary.

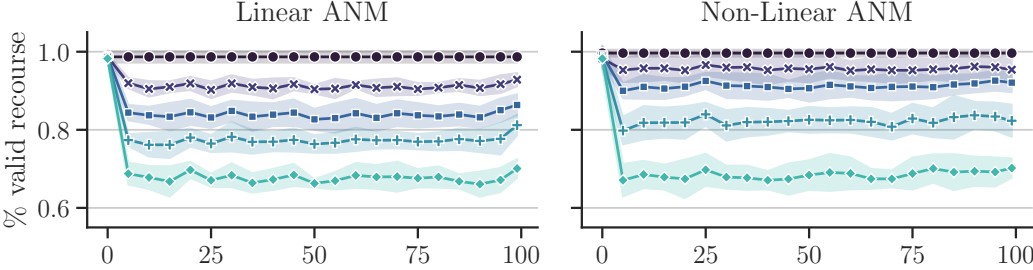

Figure 6: **Effect of uncertainty on counterfactual AR over time.** Empirical average validity and standard deviation over 10 runs of *robust* counterfactual algorithmic recourse (CAR) for $t \in \{0, 100\}$. We vary the variance $\sigma_U$ of the exogenous factors of the stochastic process. Legend ($\sigma_U$): ■ 0 ■ 0.3 ■ 0.5 ■ 0.7 ■ 1.0.

# E  ANALYSIS OF THE TRADE-OFF BETWEEN VALIDITY AND COST

We report an analysis of the trade-off between validity and cost of the recourses found by Algorithm 1. We replicate the same experimental setting and evaluation procedure as Sections 4.1 and 4.2, and we measure the effect of varying the $\lambda$ parameter controlling the strength of the recourse constraint (line 5, Algorithm 1). We consider the time series exhibiting the more complex linear+seasonal trend.

Fig. 7 show the results for the synthetic and realistic time series. The cost-validity trade-off is apparent in all the experimental settings, where **cheaper interventions yield lower validity over time**. This result complements previous findings in the literature considering non-temporal settings (Pawelczyk et al., 2022b). For example, in Adult, we observe a reduction in the validity over time ($\sim 0.05$), but a decreased cost as shown by the lighter dots. In COMPAS, the trade-off presents a smoother behaviour since we have only one actionable feature. Lastly, we observe how the reduction in validity is not consistent across time series: we suspect this might depend on the quality of the estimator $\tilde{P}(\mathbf{X}^t)$ and on the decision boundary of the classifier $h$.

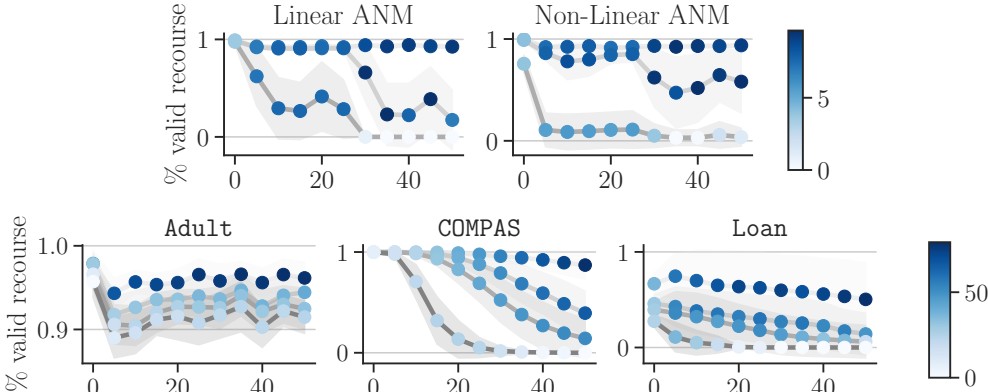

Figure 7: **Trade-off between cost and validity for realistic and synthetic datasets.** We report the empirical average validity for Algorithm 1 under the linear+seasonal trend ($\alpha = 1.0$) when varying the $\lambda$ for (Top) synthetic and (Bottom) realistic time series. We consider a non-linear classifier $h$ (3-layer neural network) for each setting. Each dot represents the empirical average cost of the interventions achieving recourse. A darker dot implies a larger cost. We represent the validity for each $\lambda$ as a grey line and the standard deviation over 10 runs as a shaded area. Legend ($\lambda$): ▢ 1.00 ▢ 0.33 ▢ 0.20 (Synthetic time series), ▢ 100 ▢ 20 ▢ 10 ▢ 2 (Realistic time series).

## F    FURTHER ANALYSIS ON THE EMPIRICAL COST AND SPARSITY

In this section, we report further analysis and results when considering the empirical average *cost* and *sparsity* (# of $\mathcal{I}$ achieving recourse) of the valid interventions. They are both common metrics in the algorithmic recourse literature (Karimi et al., 2022; Verma et al., 2020).

**Sparsity**. Figs. 8 and 9 shows the empirical average sparsity of the causal recourse methods for both synthetic and realistic stochastic processes. We do not report values for IMF since it always acts on *all* the actionable features ($|\mathcal{I}| = 3$). In Fig. 8, T-SAR presents a similar or lower sparsity than other methods. However, as Section 4.1 shows, T-SAR is the only method achieving good validity over time. Thus, these results suggest that incorporating time might not increase the sparsity of the solutions. In the case of approximate SCMs, Fig. 9 shows how T-SAR provides sparse interventions for Adult, but increasingly larger interventions for Loan. COMPAS has only one actionable feature, thus the sparsity is equal for all approaches. As highlighted in Section 4.2, T-SAR performance is also dependent on the quality of the estimator $\tilde{P}(\mathbf{X}^t)$.

**Cost.** Figs. 10 and 11 shows the empirical average cost for the users for which all (non-)causal recourse methods found a valid intervention. We consider only the top-3 methods achieving recourse for each time step $t$. In realistic and synthetic settings, T-SAR can provide *cost-adaptive* interventions which follow the underlying trend. For example, we can observe this phenomenon in both COMPAS and Loan (Fig. 11). It is also visible for $m(t) \in \{\text{Linear}, \text{Linear+Seasonal}\}$ in the Non-linear and Linear ANMs, respectively. We also notice how T-SAR seems to provide costlier recourses than the standard robust methods. However, in Adult, T-SAR produces cheaper interventions than the other approaches. We can explain this behaviour for Adult by looking at the analysis of the successful intervention sets $\mathcal{I}$ in Section 4.2.

In conclusion, by incorporating an estimator of the stochastic process, we can provide *sparse* interventions more resilient to time. These interventions tend to be costlier and the cost varies with the time lag $\tau$ when they will be applied. However, robust (non-)causal methods achieve dissimilar validity, thus making them not fully comparable to each other by measuring their cost. Nevertheless, we believe the analysis has merit since it hints at a tradeoff between sparsity, cost and validity.

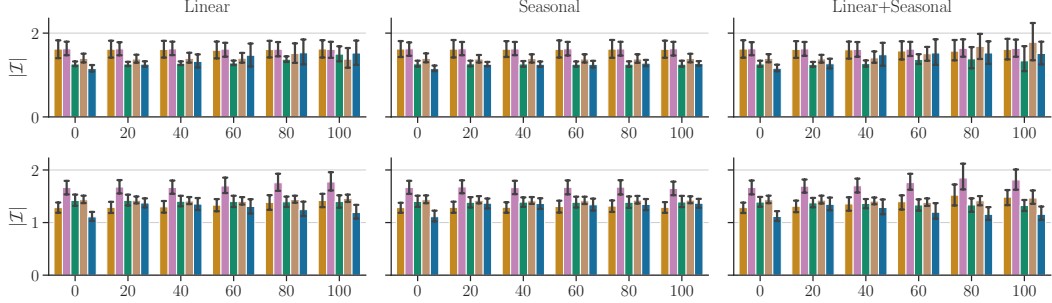

Figure 8: Empirical average sparsity and standard deviation of interventions achieving recourse for all causal recourse methods in the Linear (top) and Non-Linear (bottom) ANMs. We report the results for all the available trends $m(t) \in \{\text{Linear}, \text{Seasonal}, \text{Linear+Seasonal}\}$ and for some time steps $t$. Legend: ■ T-SAR ■ CAR ($\epsilon = 3$) ■ SAR ($\epsilon = 3$) and ■ CAR ($\epsilon = 5$) ■ SAR ($\epsilon = 5$).

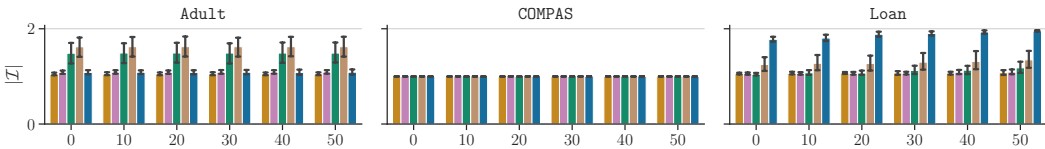

Figure 9: Empirical average sparsity and standard deviation of interventions achieving recourse for all causal recourse methods in the realistic datasets. We report the results for $m(t) = $ Linear+Seasonal and for some time steps $t$. Legend: ■ T-SAR ■ CAR ($\epsilon = 3$) ■ SAR ($\epsilon = 3$) and ■ CAR ($\epsilon = 5$) ■ SAR ($\epsilon = 5$).

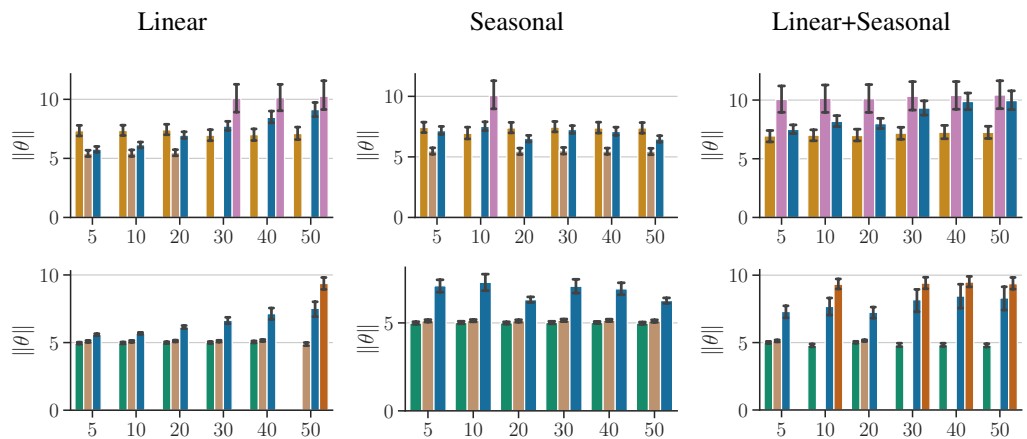

Figure 10: Empirical average cost and standard deviation for the top-3 methods achieving recourse in the Linear (top) and Non-Linear (bottom) ANMs. We report the results for all the available trends $m(t) \in \{\text{Linear, Seasonal, Linear+Seasonal}\}$. Legend: ■ T-SAR ■ CAR ($\epsilon = 3$) ■ SAR ($\epsilon = 3$) ■ IMF ($\epsilon = 3$) and ■ CAR ($\epsilon = 5$) ■ SAR ($\epsilon = 5$). ■ IMF ($\epsilon = 5$).

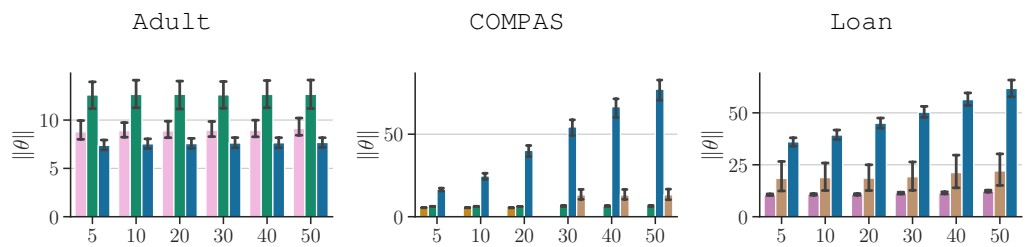

Figure 11: Empirical average cost and standard deviation for the top-3 methods achieving recourse in the realistic datasets under a non-linear trend. Legend: ■ T-SAR ■ SAR ($\epsilon = 0.05$) and ■ CAR ($\epsilon = 0.5$) ■ SAR ($\epsilon = 0.5$) ■ IMF ($\epsilon = 0.5$).

# G FURTHER EXPERIMENTS WITH A PERFECT ESTIMATOR $P(\mathbf{X}^t)$

We replicated the experiments in Section 4.2 by using instead the perfect estimator $\tilde{P}(\mathbf{X}^t) = P(\mathbf{X}^t)$ of the stochastic process for each dataset. Fig. 12 shows how T-SAR offers superior performances in terms of validity than robust (non-)causal algorithmic recourse methods. In COMPAS and Loan, T-SAR achieves now perfect validity over all time steps. These results highlight the importance of relying on a good estimator of the stochastic process and, as outlined in Section 6, we argue it is also a mandatory requirement for realistic applications of the proposed method.

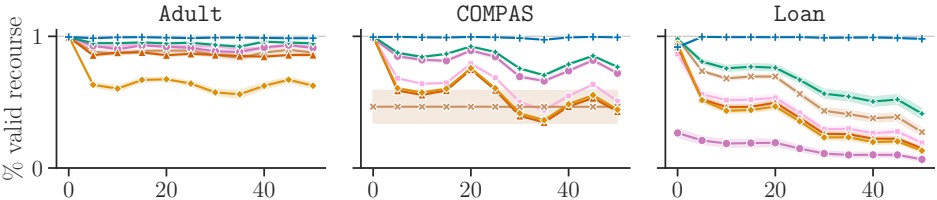

Figure 12: **Effect of time on realistic datasets.** Empirical average validity and standard error (10 runs) for the robust ($\epsilon \in \{0.05, 0.5\}$) and time-aware causal recourse methods for the realistic datasets under a non-linear trend. Legend: ■ T-SAR ■ CAR ($\epsilon = 0.05$) ■ SAR ($\epsilon = 0.05$) ■ IMF ($\epsilon = 0.05$) and ■ CAR ($\epsilon = 0.5$) ■ SAR ($\epsilon = 0.5$) ■ IMF ($\epsilon = 0.5$).

# H  ADDITIONAL THEORETICAL RESULTS ON COST STABILITY

In this section, we provide an upper bound on the cost stability of recourse suggestions when $P(\mathbf{X}, Y)$ is a discrete-time stochastic process.

Previous research has shown how providing recourse without considering the user's preferences can lead to sub-optimal interventions (De Toni et al., 2023b). This is why *personalized* AR, in line with multi-attribute decision making (Keeney & Raiffa, 1993; Pigozzi et al., 2016), models the cost function as an *additive independence model* $C(\hat{\mathbf{x}}, \mathbf{x}) = \mathbf{w}^\top |\hat{\mathbf{x}} - \mathbf{x}|$, where the weights $\mathbf{w} \in \mathbb{R}^d$ encapsulate the user's preferences (De Toni et al., 2023a;b). We assume we can learn these weights, either from historical data, *e.g.*, surveys and interviews (Rawal & Lakkaraju, 2020), or by interacting with the end-user (De Toni et al., 2023b). In the following, we explicitly consider the evolution of the user's preferences $\mathbf{W}$, as it also impacts the effectiveness of recourse, although doing so can be avoided for non-personalized AR approaches.

We assume the user preferences can be represented as a stochastic process $P(\mathbf{W}^t)$. We do not put any prior assumption on how $P(\mathbf{W}^t)$ factorizes, since it is not relevant for our results. In line with previous work (De Toni et al., 2023b), we could imagine $P(\mathbf{W}^t)$ follows a causal model. Then, we can provide the following upper bound:

**Theorem 8.** *Consider the discrete-time stochastic processes $P(\mathbf{X}^t, Y^t)$, $P(\mathbf{W}^t)$ and a parametrized cost function $C(\hat{\mathbf{x}}, \mathbf{x}; \mathbf{w}) = \langle |\hat{\mathbf{x}} - \mathbf{x}|, \mathbf{w} \rangle$ with bounded $-k \leq w_i^t, X_i^t \leq k$ for $k \in \mathbb{R}^+$. Given a realization $\mathbf{x}^t$ and user's preferences $\mathbf{w}^t$, the variation of the cost of an intervention $\boldsymbol{\theta}$ is upper bounded by:*

$$\mathbb{E}\left[|C(\hat{\mathbf{x}}^{t+\tau}, \mathbf{x}^{t+\tau}; \mathbf{w}^{t+\tau}) - C(\hat{\mathbf{x}}^t, \mathbf{x}^t; \mathbf{w}^t)|\right]$$
$$\leq k\sqrt{d} \cdot \mathbb{E}\left[\|\mathbf{w}^{t+\tau} - \mathbf{w}^t\| + \||\hat{\mathbf{x}}^{t+\tau} - \mathbf{x}^{t+\tau}| - |\hat{\mathbf{x}}^t - \mathbf{x}^t|\|\right] \tag{31}$$

*where $\hat{\mathbf{x}}^t \sim P^{do(\boldsymbol{\theta})}(\mathbf{X}^t \mid \mathbf{X}_{nd(\mathcal{I})}^t = \mathbf{x}_{nd(\mathcal{I})}^t)$ and $\hat{\mathbf{x}}^{t+\tau} \sim P^{do(\boldsymbol{\theta})}(\mathbf{X}^{t+\tau} \mid \mathbf{X}_{nd(\mathcal{I})}^{t+\tau} = \mathbf{x}_{nd(\mathcal{I})}^{t+\tau})$.*

Theorem 8 shows how the recourse cost changes based on how the users' preferences evolve, and also over the relative difference of the proposed changes given the starting value.

*Proof.* We first apply the following substitutions (a) $\mathbf{w}' = \mathbf{w}^{t+\tau}$ and $\hat{\mathbf{x}}' = \hat{\mathbf{x}}^{t+\tau}$ (b) $\mathbf{x}' = \mathbf{x}^{t+\tau}$ $\hat{\mathbf{x}}'' = \hat{\mathbf{x}}^t$ (c) $\mathbf{w} = \mathbf{w}^t$ and $\mathbf{x}'' = \mathbf{x}^t$, to improve the clarity of the proof. Then, the proof is the following:

$$\mathbb{E}\left[|C(\mathbf{x}, \mathbf{x}; \mathbf{w}') - C(\mathbf{x}, \mathbf{x}; \mathbf{w})|\right] = \mathbb{E}\left[|\langle |\mathbf{x}' - \mathbf{x}|, \mathbf{w}' \rangle - \langle |\mathbf{x} - \mathbf{x}|, \mathbf{w} \rangle|\right]$$
$$= \mathbb{E}\left[|\langle |\hat{\mathbf{x}}' - \mathbf{x}'| - |\hat{\mathbf{x}}'' - \mathbf{x}''|, \mathbf{w}' \rangle + \langle |\mathbf{w}' - \mathbf{w}|, |\hat{\mathbf{x}}'' - \mathbf{x}''| \rangle|\right]$$
$$\overset{(i)}{\leq} \mathbb{E}\left[\||\hat{\mathbf{x}}' - \mathbf{x}'| - |\hat{\mathbf{x}}'' - \mathbf{x}''|\| \cdot \|\mathbf{w}'\|\right] + \mathbb{E}\left[\||\mathbf{w}' - \mathbf{w}|\| \cdot \||\hat{\mathbf{x}}'' - \mathbf{x}''|\|\right]$$
$$\overset{(ii)}{\leq} k\sqrt{d} \cdot \mathbb{E}\left[\||\hat{\mathbf{x}}' - \mathbf{x}'| - |\hat{\mathbf{x}}'' - \mathbf{x}''|\|\right] + k\sqrt{d} \cdot \mathbb{E}\left[\||\mathbf{w}' - \mathbf{w}|\|\right]$$
$$= k\sqrt{d} \cdot \mathbb{E}\left[\||\hat{\mathbf{x}}' - \mathbf{x}'| - |\hat{\mathbf{x}}'' - \mathbf{x}''|\| + \||\mathbf{w}' - \mathbf{w}|\|\right]$$
$$= k\sqrt{d} \cdot \mathbb{E}\left[\|\mathbf{w}^{t+\tau} - \mathbf{w}^t\| + \||\hat{\mathbf{x}}^{t+\tau} - \mathbf{x}^{t+\tau}| - |\hat{\mathbf{x}}^t - \mathbf{x}^t|\|\right]$$

$$\tag{32}$$

where (*i*) follows from the Cauchy-Schwarz inequality, and (*ii*) from the bounds we placed on $\mathbf{X}$ and $\mathbf{W}$. On the last step, we reorder the terms and we substitute the temporary variables with the original values. $\square$

