# OpenReview forum: "Time Can Invalidate Algorithmic Recourse"
_ICLR.cc/2025/Conference — Submitted to ICLR 2025_

### Official Review · Reviewer_g3T6 · 2024-10-28

**Soundness:** 3
**Presentation:** 3
**Contribution:** 3
**Rating:** 8
**Confidence:** 4

**Summary:**

The authors argue that some features based on which decision are made might change over time and theoretically study the robustness of algorithmic recourse in this context. They find that algorithmic recourse can be invalidated by non-stationary and exogeneous noise over time. Upon their theoretical findings, the paper proposes a novel algorithm that explicitly incorporates time in the recourse computation and empirically shows that the proposed algorithm suggests recourses with higher validity over time.

**Strengths:**

- the paper is well-written and easy to follow
- the theoretical insights are novel and important for practical applications of algorithmic recourse
- proofs and derivations are complete and correct
- the empirical evaluation demonstrates the advantages of the proposed algorithm incorporating time explicitly

**Weaknesses:**

- Def. 1 could be discussed a bit more in detail. It is not immediately clear what $P$ and $Q$ refer to in the definition. Although it is mentioned a bit further below, it would increase the clarity of the paper if $Q$ and $P$ are briefly described in Def. 1.
- Eq. 5: It is unclear how $\theta$ and $\Delta$ are related. Since $\Delta$ denotes a set of interventions, the notation is not intuitive. The authors should describe their relationship in a brief sentence.
- in lines 338-340, the authors write: "The upper bound is also useful for non-linear classifiers $h$ since we can represent their decision function close to a realization $\mathbf{x}^t$ with a local linear approximation." While I agree that non-linear functions can be approximated using a set of linear functions, it is not immediately clear why the upper bound from Theorem 6 should be useful for non-linear approximators in general. Having a set of linear functions approximating some non-linear function requires somehow merging/combining the upper bounds of the linear functions. Doing this is generally non-trivial since each linear function only considers a local subspace; thus, straightforward solutions like taking the $\max$ of all upper bounds would not work in general to obtain a global upper bound, would it?
- Fig. 3: I agree that the uncertainty clearly has a negative effect on validity. However, one cannot directly see the effect of time in this plot. An additional time-specific plot would be beneficial to underline the statement made in Fig. 3.
- Experimental Setup (4.2): A brief discussion on how the structure of the approximate SCM was obtained, as well as on how good the approximations of the structural equations were, would be beneficial.

**Questions:**

- Def. 4: Is $P(Y^t | \mathbf{X}^t)$ assumed to be stationary here (since it was mentioned that only covariate-shift-like distribution shifts are considered in Sec. 3.1)? Because if so, distinguishing different $h^t$ would not make sense in my opinion.
- line 364: Why could the proposed algorithm not be used for CAR?
- Fig. 5: Why is the standard deviation so high on Loan for T-SAR?

---

> ### Author Response · Authors · 2024-11-18
> **Answer to Reviewer g3T6**
>
> 1. > __it would increase the clarity of the paper if Q and P are briefly described in Def. 1__
>
> We revised Definition 1 to include the potential choices of Q within the definition.
>
> 2. > __Eq. 5: It is unclear how θ and Δ are related__
>
> We revised Definition 3 to describe $\Delta$ as “perturbations” instead of interventions to underline their distinction from $\theta$.
>
> 3. > __While I agree that non-linear functions can be approximated using a set of linear functions, it is not immediately clear why the upper bound from Theorem 6 should be useful for non-linear approximators in general__
>
> By a linear approximation of a non-linear model we mean simple models such as LIME [1] which provide a local linear approximation of the decision boundary near a given point $\mathbf{x}$ by training a single linear model $h(\mathbf{x})$. Following the reviewer's suggestion, we clarified the statement in the revised manuscript.
>
> 4. > __An additional time-specific plot would be beneficial to underline the statement made in Fig. 3.__
>
> As suggested by the reviewer, we added a time-specific plot in Appendix D and referenced it in the main manuscript (line 429).
>
> 5. > __Experimental setup: A brief discussion on how the structure of the approximate SCM was obtained, as well as on how good the approximations of the structural equations were, would be beneficial.__
>
> We describe the structure of the approximate SCM and additional details in Appendix C.4 and C.5. In practice, following previous approaches [2], we learn a separate generative model for each structural equation of the SCM and we train it. We assume we can represent the actionable features $X_i$ as Gaussian random variables $\mathcal{N}(\mu^t_i, 1)$ with constant variance and time-dependent $\mu^t$. For each random variable $X_i$, we define the mean as the output of a regressor $f_i$ taking as input the parents $\mathbf{Pa}^t_{X_i}$ and the time $t$. We train each regressor by minimizing the MSE over a batch of observations over the corresponding random variable $X_i$. We report here the MSE over 50 timesteps (2000 individuals) for the approximate SCMs we used in Section 4.2. We compute the MSE for each feature for each timestep, and then we average. We show the empirical average MSE and standard deviation over 10 runs: $1.162 \pm 0.005$ (Adult), $258.226 \pm 7.328$ (COMPAS) and $10.447 \pm 0.037$ (Loan). We revised Appendix C.5 to add these results.
>
> 6. > __Def. 4: Is P(Yt|Xt)  assumed to be stationary here__
>
> No, we assume $P(Y^t \mid \mathbf{X}^t)$ is not stationary in Definition 4. We clarified this point in the revised version of the manuscript (lines 304-305).
>
> 7. > __line 364: Why could the proposed algorithm not be used for CAR?__
>
> In practice, the reviewer is right and Algorithm 1 could also be used to optimise CAR over time, rather than SAR. However, in Section 3.2, we showed the optimum for CAR cannot be recovered when the exogenous variables have non-zero variance, which happens in the vast majority of situations. Thus, we felt that optimising for SAR was more appropriate because this setup – following the relevant literature [2] - is considered more easily implementable and realistic.
>
> 8. > __Fig. 5: Why is the standard deviation so high on Loan for T-SAR?__
>
> Loan represents a non-linear SCM (as described in Appendix C.3) and we empirically found it was the most difficult to optimize for. Our suggested recourses are still robust to time in expectation but since our estimator of $P(\mathbf{X}^t)$ is not perfect, they exhibit a larger standard deviation. Indeed, in the experiments with a perfect estimator (Appendix G), the standard deviation is much smaller.
>
> [1] Ribeiro et al. "" Why should I trust you?" Explaining the predictions of any classifier." SIGKDD (2016)
>
> [2] Karimi et al. "Algorithmic recourse under imperfect causal knowledge: a probabilistic approach." NeurIPS (2020)

---

> > ### Comment · Reviewer_g3T6 · 2024-11-20
> > **Response to Rebuttal**
> >
> > I thank the authors for their response to my review. All my concerns & questions have been addressed and answered.
> >
> > After reading the other reviews and the corresponding rebuttals, I will keep my score (see comment below).
> >
> > Also, I'd like to briefly comment on the doubts about the practicality of the proposed approach raised by reviewers YDw8 and Tspd:
> > While I acknowledge that finding an accurate estimator $P(\mathbf{X}^t)$ can be challenging, I would not see this as a major limitation of the proposed approach incorporating time in AR. As the authors have highlighted in the response to reviewer YDw8, even imperfect estimators help already.
> >
> > Considering that this work seems to be the first explicitly incorporating time (lags) in AR and considering the important theoretical and promising empirical results provided in the paper, I think this work is worth publishing.
> >
> > Since there is no rating option 9 and I think that, for 10, the paper is not strong enough, I kindly ask the ACs to view my rating as a 9.

---

### Official Review · Reviewer_L6ZZ · 2024-10-30

**Soundness:** 2
**Presentation:** 2
**Contribution:** 3
**Rating:** 5
**Confidence:** 4

**Summary:**

This paper studies the problem of algorithmic recourse in the time series setting. Previous works on algorithmic recourse neglect the impact of time evolution on the variables. The authors theoretically analyze the validity of original algorithmic recourse in non-stationary and uncertain world and the stability of recourse over time. Based on the theroretical analysis, the authors propose a simple yet effective algorithm to produce the AR that can counteract the time. The experimental results on synthetic datasets and real-world datasets shows that the AR generated by the proposed method (T-SAR) can remain valid across time steps.

**Strengths:**

This paper investigate an important but not studied problem, that is the influence of time on algorithmic recourse. This paper explore the problem from the pespective of both theory and empirical examinations. Extensive theoretical analysis is sound and comprehensive. The empirical improvement in experiements is significant, showing the effectiveness of the proposed method.

**Weaknesses:**

1. The presentation of the paper could be further improved. The problem formulation is not clear and kind of vague. I guess the problem is to determine the intervention $\theta$ at time $t$ and implement it at a later time $t+\tau$ instead of determining intervention at $t+\tau$. However, it is not distinguished in the manuscript. And the notation $do(\theta)$ is confusing, because it is not clear that at which time step the intervention $\theta$ is conducted.
2. I think in the field of algorithmic recourse, the key component about label is the classifer $h$. After $h$ is trained, the true label makes no effect on the generation of AR, since the AR is computed to reverse the predicted label instead of true label. I guess it is needed to adjust the statement in the theoretical analysis.

**Questions:**

1. I think some notations are confusing. For example, in E.q.(9) the $B(\mathbf{x}^t;\tau)$ refers to the future feature after time $\tau$. However, in line 388, the example of $B(\mathbf{x}^t;\tau)$ is not relevant to time interval $\tau$.
2. I want the authors to explain the sentence "However, this does not prevent invalidation if the user implements this updated
recourse later on." in line 208.
3. I want to know how the causality play the role in the proposed method.
4. Did the authors examine whether the produced recourse is compatible with the causal relationship among variables.

---

> ### Author Response · Authors · 2024-11-18
> **Answer to Reviewer L6ZZ**
>
> 1. > __The problem formulation is not clear and kind of vague. [...]. And the notation do(\theta) is confusing, because it is not clear that at which time step the intervention is conducted__
>
> Given the current state of the user $x^t$, Definition 1 considers the problem where we want to find the recourse $\theta$ which, **if applied at time $t+\tau$** would achieve recourse at $t+\tau$. Following the reviewer’s suggestion, we revised Definition (1) to state more clearly the problem statement and we added a clarification to the $do(\theta)$ notation (lines 207-208) about the time when the intervention $\theta$ is applied.
>
> 2. > __[...] the key component about label is the classifer h. After h is trained, the true label makes no effect on the generation of AR. [...] I guess it is needed to adjust the statement in the theoretical analysis.__
>
> We would like to clarify how our theoretical analysis **always** considers a classifier $h$ approximating the conditional $P(Y \mid \mathbf{X})$ in line with the recourse literature. The only exception is Proposition 1 since it concerns the feasibility of computing the counterfactual distribution over the future from a causal perspective. We believe the confusion stemmed from us reporting the joint distribution $P(\mathbf{X}^t, Y^t)$, rather than simply $P(\mathbf{X}^t)$. We corrected this in the revised manuscript by updating all the theorems, propositions and corollaries.
>
> 3. > __I think some notations are confusing. For example, in E.q.(9) the B(x^t;τ) refers to the future feature after time τ. However, in line 388, the example of B(x^t;τ) is not relevant to time interval τ.__
>
> We revised the notation in the manuscript by removing the dependence over $\tau$ for the uncertainty set $B$ defined for non-causal recourse (page 8, lines 388-389).
>
> 4. > __I want the authors to explain the sentence "However, this does not prevent invalidation if the user implements this updated recourse later on." in line 208.__
>
> This naive solution would be to wait until the user reaches $t+\tau$, so that we can observe her new state $\mathbf{x}^{t+\tau}$, and then compute a new recourse suggestion. However, in our setting, the user could still implement this new suggestion at a later time $t+\tau’$, where $\tau’ > \tau$ and she would not be guaranteed to obtain recourse.
>
> 5. > __[...] how the causality play the role in the proposed method.__
>
> In our work, we assume the stochastic process $P(\mathbf{X}^t, Y^t)$ factorizes following a causal structure represented by a structural causal model (SCM) [1]. We define a recourse suggestion as a causal soft intervention $do(\mathbf{X}=\mathbf{x}+\theta)$. Thus, our recourse problem boils down to finding the cheapest intervention $\theta$ such that the user will obtain a positive classification (in expectation), as per Definition 1. Moreover, given a recourse intervention $\theta$, the SCM enables us to compute the causal effects it produces (in expectation) on the user’s features once it is applied e.g., via the interventional or counterfactual distributions (Eq. 2). Lastly, the SCM enables us to simulate the user evolution over time to compute the uncertainty set B (Equation 9, line 371). All these aspects are used by Algorithm 1 to generate recourse suggestions robust to time by exploiting the causal structure of the stochastic process **by design**.
>
> 6. > __Did the authors examine whether the produced recourse is compatible with the causal relationship among variables.__
>
> Following the answer for _(5.)_, we remark that Algorithm 1 produces recourse interventions satisfying the causal relationship among variables **by design**. This also includes considering their potential non-actionability, as noted in footnote 1. E.g., we cannot ask someone to get younger as an intervention.
>
> [1] Pearl, Judea. Causality. Cambridge University Press, 2009.

---

> > ### Author Response · Authors · 2024-11-22
> >
> > Dear Reviewer,
> >
> > We would like to follow up to see if our response addresses your concerns or if you have further questions. We would really appreciate the opportunity to discuss this further and know whether our response has already addressed your concerns. Thank you again!

---

> > ### Comment · Reviewer_L6ZZ · 2024-11-26
> >
> > I thank the authors for their replies.
> >
> > I still have mains concerns about Question 3. How to construct the a structural causal model is not clear in this paper. Therefore, the causality constraints on $\theta$ has not been elborated. The Algorithm 1 does not show the \textbf{design} of exploiting the causal structure.
> >
> > So, I decide to maintain my score.

---

> > > ### Author Response · Authors · 2024-11-26
> > >
> > > > **I still have mains concerns about Question 3. How to construct the a structural causal model is not clear in this paper. Therefore, the causality constraints on has not been elborated.**
> > >
> > > We would like to clarify further where we elaborate on the causality in our paper:
> > >
> > > - We introduce our causal model (TiMINO) in **Definition 2**.
> > > - Its role in Temporal Algorithmic Recourse is outlined in **Definition 1**: the causal model yields the distribution $Q$ used for assessing the probability of achieving recourse (the objective) and the expected cost of said recourse (the constraint) (**lines 176, 177**).
> > > - We have discussed in **lines 191-196** how TiMINos can be learned from (even observational) data under appropriate assumptions.
> > > - We describe in Appendix C the structural causal models for the experiments and in Appendix C.4 and C.5 **how we construct the structural causal models** for the real-world experiments in Section 4.2.
> > > - We have updated the manuscript to underline the link between the robustness set $B$ and the TiMINo causal model at **line 368**.
> > >
> > > Please let us know if anything is still unclear and how you suggest we improve the manuscript.
> > >
> > > > **The Algorithm 1 does not show the \textbf{design} of exploiting the causal structure.**
> > >
> > > Our algorithm is an extension of existing causal AR methods [1,2,3,4,5]. It uses the causal graph in the same way as those methods to evaluate the probability and expected cost of recourse to select high-quality, actionable suggestions for users to implement. For example, **the derivative on line 6 is computed by exploiting the interventional effect of $\theta$ on the causal graph** (in practice, it is implemented by automatic differentiation on the SCM structure as in other works such as [2]).
> > >
> > > > **So, I decide to maintain my score.**
> > >
> > > We disagree that this is grounds for rejection. Our contributions focus on studying theoretically and empirically a new phenomenon hindering the robustness of recourse, while the reviewer’s critiques focus on the **widely accepted standard formalization of causal algorithmic recourse** in the literature (e.g., [1,2,3,4,5]) that we **follow**.
> > >
> > > Moreover, **our empirical results show that non-causal recourse is also fragile to time**, see Figures 2, 3, 4 and 5. This has immediate, significant consequences for all commonly used algorithmic recourse solutions.
> > >
> > > We would appreciate it if you could engage in the discussion, we will be glad to iron out any remaining issues you might point out.
> > >
> > > [1] Karimi et al. "Algorithmic recourse under imperfect causal knowledge: a probabilistic approach." NeurIPS (2020)
> > >
> > > [2] Dominguez-Olmedo et al. "On the adversarial robustness of causal algorithmic recourse." ICML (2022)
> > >
> > > [3] Von Kügelgen et al. "On the fairness of causal algorithmic recourse." AAAI (2022)
> > >
> > > [4] Ehyaei et al. "Robustness implies fairness in causal algorithmic recourse.” FAccT (2023)
> > >
> > > [5] Majumdar and Valera. "CARMA: A practical framework to generate recommendations for causal algorithmic recourse at scale." FAccT (2024)

---

### Official Review · Reviewer_Tspd · 2024-10-31

**Soundness:** 3
**Presentation:** 3
**Contribution:** 2
**Rating:** 5
**Confidence:** 3

**Summary:**

This work investigates the temporal robustness of Algorithmic Recourse (AR), focusing on how AR methods often lose validity over time due to non-stationary data environments. The authors propose a time-aware causal recourse approach, "Temporal Sub-population Algorithmic Recourse (T-SAR)," designed to adapt to dynamic conditions by forecasting recourse stability. They theoretically and empirically demonstrate that both conventional and robust AR techniques struggle to maintain relevance over time, especially in non-deterministic environments, where optimal solutions are typically unattainable. The proposed T-SAR method shows promising results in maintaining recourse validity under various temporal trends and uncertainties.

**Strengths:**

The concept of Temporal Sub-population Algorithmic Recourse (T-SAR) adds a unique dimension by incorporating time-awareness, which is relatively novel in the field of causal AR. The theoretical contributions are well-grounded in causal inference, and the authors provide a robust theoretical framework that explains why existing AR models may fail over time. By addressing the challenges of AR in non-static environments, this work highlights an important area for improvement in AR techniques. The emphasis on time-sensitive solutions could stimulate further research in ensuring AR methods remain effective as conditions change.

**Weaknesses:**

The explanation of how temporal sensitivity interacts with causality is unclear. For instance, while the paper mentions the impact of time on causal interventions, it doesn’t sufficiently explain how different timing of interventions might lead to varied results, especially in non-stationary environments. This leaves readers uncertain about the long-term implications of time on AR effectiveness, as the dynamic causal effects of time-sensitive actions are only superficially addressed.
The paper assumes precise trend estimation, which is unrealistic in many real-world applications where trend data is noisy and unreliable. This assumption affects T-SAR’s practical applicability, as trend inaccuracies could lead to degraded performance. Although the paper mentions non-stationary conditions, it lacks a thorough discussion of the limitations that could arise if the trend estimator is flawed.

**Questions:**

1.	Proposition 1 states that achieving optimal counterfactual recourse depends on zero variance in exogenous factors.  In cases where variance is unavoidable, how significant is the impact on the effectiveness of T-SAR recommendations? Could you quantify this impact and discuss alternative methods to ensure robust recourse.
2.	Proposition 3 claims that optimal recourse remains valid as long as the stochastic process is stationary. However, real-world processes rarely exhibit perfect stationarity. Could you discuss specific non-stationary scenarios where T-SAR might underperform? Are there any adaptive features in T-SAR to account for non-stationary processes, or must stationarity be a prerequisite for its application?

---

> ### Author Response · Authors · 2024-11-18
> **Answer to Reviewer Tspd**
>
> 1. > __The explanation of how temporal sensitivity interacts with causality is unclear.__
>   > __[The paper] doesn’t sufficiently explain how different timing of interventions might lead to varied results, especially in non-stationary environments.__
>
> We would like to clarify that we consider a setting where any recourse is implemented, and its total causal effects are observed, at time $t + \tau$, and $\tau > 0$ is fixed. Therefore, the timing $\tau$ when the user chooses to perform the intervention is crucial to determine the validity of the recourse. We remark that in this setting, time hinders the validity of recourse because of the potential non-stationarity in the stochastic process, producing diverse causal effects depending on $\tau$. Following the reviewer’s comment, we revised Definition 1 and “Limitations” to better underline our modelling choice and the dynamic of the actions.
>
> 2. > __The paper assumes precise trend estimation [...]. [This] affects T-SAR’s practical applicability.__
>   > __[The paper] lacks a thorough discussion of the limitations that could arise if the trend estimator is flawed.__
>
> We wish to remark how, in Section 4.2, we show empirical evidence on realistic datasets that an imperfect estimator of $P(\mathbf{X}^t)$ can still be beneficial. We agree that estimating $P(\mathbf{X}^t)$ can be hard, and we revised the “Limitations” (Section 5) to better state our assumptions and the underlying difficulty of learning a good estimator.
>
> 3. > __Proposition 1 states that achieving optimal counterfactual recourse depends on zero variance in exogenous factors. In cases where variance is unavoidable, how significant is the impact on the effectiveness of T-SAR recommendations? Could you quantify this impact and discuss alternative methods to ensure robust recourse.__
>
> We would like to clarify that Algorithm 1 solves the _temporal subpopulation recourse problem_ (T-SAR) since it considers the interventional distribution in Definition 1, rather than the counterfactual one. This fact enables us to sidestep the negative results of Proposition 1. Moreover, Algorithm 1 robustifies **by design** against the uncertainty coming from the exogenous factors by returning an intervention $\theta$ valid within an uncertainty set $B(\mathbf{x}^t; \tau)$ (Equation 9, Section 3.6). We construct the uncertainty set by sampling a finite number of individuals from the estimator $\tilde{P}(\mathbf{X}^{t+\tau} \mid \mathbf{X}^t = \mathbf{x}^t)$ conditioned on the current state of the individual $\mathbf{x}^t$. In practice, we are simulating the evolution of the user until $t+\tau$ by considering a “set” of likely future versions of the user, and we ensure the recourse suggestion is valid for all of them. Since the data manifold generated by the uncertainty set $B(\mathbf{x}^t; \tau)$ conforms to the causal generative process (under the assumption of having a reasonable estimator), we are effectively taking into account and counteracting the uncertainty coming from the exogenous factors when computing recourse.
>
> 4. > __Proposition 3 claims that optimal recourse remains valid as long as the stochastic process is stationary. However, real-world processes rarely exhibit perfect stationarity. Could you discuss specific non-stationary scenarios where T-SAR might underperform? Are there any adaptive features in T-SAR to account for non-stationary processes, or must stationarity be a prerequisite for its application? discuss alternative methods to ensure robust recourse__
>
> In our work, we account for the non-stationarity of the process by giving Algorithm 1 access to an estimator $\tilde{P}(\mathbf{X}^t)$ such to provide robust recourse suggestions accounting for both stationary and non-stationary time series. As described in _(3.)_, this feature enables Algorithm 1 to provide **by design** robust recourse solving T-SAR. Moreover, we would like to point the reviewer to Sections 4.1 and 4.2 where we provide an empirical evaluation showing that Algorithm 1 effectively provides robust recourse under several synthetic and realistic **non-stationary time series**.

---

> > ### Author Response · Authors · 2024-11-22
> >
> > Dear Reviewer,
> >
> > We would like to follow up to see if our response addresses your concerns or if you have further questions. We would really appreciate the opportunity to discuss this further and know whether our response has already addressed your concerns. Thank you again!

---

> > > ### Author Response · Authors · 2024-12-01
> > >
> > > Dear Reviewer,
> > >
> > > As we approach the end of the discussion period, we want to follow up to ensure that our previous responses have fully addressed all your questions and concerns. Please do not hesitate to let us know if you have any further questions or unresolved issues that we can clarify. Thank you again!

---

### Official Review · Reviewer_YDw8 · 2024-11-04

**Soundness:** 2
**Presentation:** 2
**Contribution:** 2
**Rating:** 3
**Confidence:** 4

**Summary:**

The paper raises an interesting point: because it takes time to implement the recourse, the distributions of the features may shift. Thus, in the future, after the recourse has been correctly implemented, the new feature is still possible to get invalidated by the classifier.

To resolve this problem, the paper proposes to estimate the future distribution of the features, and then solve a min-max iterative algorithm to find the recourse (Algorithm 1).

**Strengths:**

1. The paper studies an interesting phenomenon: the temporal dimension of algorithmic recourse. It is a relevant problem since implementing recourses can take a long time. Thus, we need to have a mechanism to take the time-horizon (in this paper is $\tau$) into the recourse problem.

**Weaknesses:**

The main downsides of this approach are

1. It requires an estimator $\tilde P(X^t)$. I would argue that in most practical settings, this distribution is never available. If it is available, it could be easily misspecified as well.

I believe there is a big gap between what the authors think are practical, and what I think are practical. It is extremely unclear to me how I could use the proposed method in a real-world deployment. The problem is complex when time is taken into account, and having access to a stochastic model $\tilde P$ in my opinion is the most difficult barrier.

2. It omits the feedback loop: the robust recourse proposed in this paper will lead to a time-horizon of interest longer than $\tau$. To see this, suppose that a non-robust recourse recommends the loan applicant to get a Master’s degree. The robust recourse will recommend a larger cost action such as to get a PhD degree. However, getting a PhD degree takes even more time, and one needs to stretch the time horizon of interest longer. Nevertheless, this feedback effect is not taken into account in this paper.

3. Algorithm 1 is a robust approach where $x$ can take any possible values in the support of $\tilde P(X^{t + \tau} | X^t = x^t)$, which can be really conservative. This can be reflected by the high costs reported in Appendix C. The authors should conduct a multi-objective analysis, for example, by studying the trade-off of cost-validity.

Some minor concerns:

1. Regarding the exposition, the paper constantly switches between the sub-population recourse and the counterfactual recourse, making it a bit hard to follow. It would be better if the authors can separate the two cases and treat them in a clear, explicit manner. This will help the readers to distinguish the characteristics of the two cases.

2. The authors even forget to state that the stochastic process should be stationary in Proposition 3.

3. The last paragraph on page 4: ``However, this does not prevent invalidation if the user implements this updated recourse later on.” I do not fully understand this argument. And on the contrary, I believe that the naive approach seems to work best! With the complexity of the problem, a simple “recompute” with proper recalibration may be the best solution. The authors should (at the very least) benchmark against this naive solution.

4. Should the right-hand side of equation (7) and (8) be smaller than 1? Could the authors please provide explicit conditions for these bounds to be meaningful?

**Questions:**

I hope the authors can address some of the weaknesses highlighted above

---

> ### Author Response · Authors · 2024-11-18
> **Answer to Reviewer YDw8**
>
> 1. > __It requires an estimator that [...] is never available [and] could be misspecified.__
>
> We agree that estimating $P(\mathbf{X}^t)$ can be difficult, but we disagree that doing so is hopeless. The time series literature keeps proposing new techniques (some more successful than others [1]) and validating them in multiple scenarios, such as the M4 competition [2], and its successive editions, where they benchmark 61 distinct methods on 100000 different time series.  Moreover, our experiments show how **even imperfect estimators can be beneficial in some settings (Section 4.2)**. But even if such an estimator is impossible to obtain, **all our negative results -- which are the core message of our work -- still apply**: time compromises the validity of recourse, which can have serious consequences.
>
> We understand the reviewer's concerns, so we updated the Abstract and “Limitations” to clarify our assumptions and the underlying difficulty of learning an estimator.
>
> 2. > __It omits the feedback loop__
>
> We would like to clarify that we consider a setting where any recourse is implemented, and its total causal effects observed, at time $t + \tau$, and $\tau > 0$ is fixed. In causal terms, we assume our SCM exhibits _instantaneous effects_ [5]. Therefore, **we do not need to extend the $\tau$ interval**, and the only uncertainty lies in which $\tau$ the user will choose to perform the intervention. We remark that even in this setting, time still hinders the validity of recourse because of the potential trends in the stochastic process, and modelling delayed causal effects would be a nice extension, but it would not invalidate our theoretical analysis. We do agree with the reviewer that, if the total causal effects of recourse cannot be observed within $t+\tau$, we would need to adaptively adjust $\tau$ in Algorithm 1 by looking at when the last causal effect will take place.
>
> Following the reviewer's concern, we revised Definition 1 and “Limitations” to better underline our modelling choice.
>
> 3. > __The authors should [study] the trade-off of cost-validity.__
>
> Following the reviewer’s suggestion, we performed **additional experiments to study the trade-off between cost and validity** by varying the $\lambda$ parameter in Algorithm 1 controlling the penalty for not achieving recourse. We report the results in Appendix E. These illustrate a trade-off between the cost of an intervention and its validity over time: **cheaper interventions almost surely have reduced validity over time.** Therefore, T-SAR suggests more conservative actions: this is sensible because what matters for users is overturning the classification e.g., being granted the loan. However, the magnitude of the reduction in validity depends on the setting, classifier $h$ and quality of the estimator.
>
> 4. > __separate sub-population and counterfactual recourse in exposition__
>
> Following the reviewer's comment, we added a table in Appendix A (to be moved to the main text later on, for visibility) summarizing the differences between all the various methods we considered, referencing our theoretical results. We also reference this table in Sec. 3 (lines 153-154) of the main text.
>
> 5. > __missing stationary in Prop 3__
>
> Thank you, we fixed the statement.
>
> 6. > __p4: ``However, this does not prevent invalidation if the user implements this updated recourse later on.”__
>
> Following our previous clarification on our setting in _(2.)_, we remark the naive solution would be to wait until the user reaches $t+\tau$, so that we can observe her new state $\mathbf{x}^{t+\tau}$, and then compute the new recourse suggestion. However, in our setting, the user could still implement this new suggestion at a later time $t+\tau’$, where $\tau’ > \tau$ and she would not be guaranteed to obtain recourse.
>
> 7. > __Should the RHS of Eqs (7) and (8) be $< 1$?  [When] are these bounds meaningful?__
>
> Eqs (7) and (8) consider a linear classifier $h(\mathbf{x}) = \langle \mathbf{x}, \beta \rangle$. Thus, the difference between $h^t$ and $h^{t+\tau}$ can be greater than 1 depending on whether $\mathbf{X}^t$ and $\beta$ are normalized in a $k$-ball. The bound can be meaningful to compare a-priori two interventions $\theta_1$ and $\theta_2$ to understand their “risk” to be invalidated before suggesting them to the user e.g., a time-agnostic intervention might cause a larger $\Delta(h, \theta)$.
>
> [1] Lim et al. "Temporal fusion transformers for interpretable multi-horizon time series forecasting" International Journal of Forecasting (2021)
>
> [2] Makridakis et al. "The M4 Competition: 100,000 time series and 61 forecasting methods" International Journal of Forecasting (2020)
>
> [3] Karimi et al. "Algorithmic recourse under imperfect causal knowledge: a probabilistic approach" NeurIPS (2020)
>
> [4] Dominguez-Olmedo et al., "On the adversarial robustness of causal algorithmic recourse" ICML (2022)
>
> [5] Peters et al., Elements of causal inference: foundations and learning algorithms (2017)

---

> > ### Author Response · Authors · 2024-11-22
> >
> > Dear Reviewer,
> >
> > We would like to follow up to see if our response addresses your concerns or if you have further questions. We would really appreciate the opportunity to discuss this further and know whether our response has already addressed your concerns. Thank you again!

---

> ### Comment · Reviewer_YDw8 · 2024-11-24
> **Answer**
>
> I thank the authors for their replies.
>
> 1. I do not doubt that our field is making better prediction models. But the authors cite M4 competition, which has nothing to do with *human* data. Recourse problems are applied to consequential domains where human lives are at stake. Unfortunately, in these domains, data are quite scarce. There is a big gap here between what the authors need (prediction model for human behavior) and the availability of data for a practical and relevant application.
>
> Time can invalidate recourses is also not something too insightful, in my opinion. This is just a simple "robustness" claim: the solution of an optimization problem is not robust when the underlying input changes. Our field has been talking about robustness for decades.
>
> 3. Could the authors explain why varying lambda is meaningful? My concern is with line 4 of Algorithm 1, where x* is already very conservative. It has nothing to do with lambda in line 5.
>
> 6. I don't understand this argument. Why don't we wait until time $t+\tau'$ and recommend another recourse when we reach time $t + \tau'$? In fact, we should first ask the person: "Are you ready to implement the recourse *now*?" If the answer is Yes, we compute the recourse. If the answer is No, then we respond: "Here is an example of recourse if you can implement now. But come back later for a more appropriate recourse when you are ready to implement" <I am just state roughly the idea, please do not quote>

---

> > ### Author Response · Authors · 2024-11-25
> > **Answer to Reviewer YDw8 (1/2)**
> >
> > > **[...] human data [...] Unfortunately, in these domains, data are quite scarce. [...]**
> >
> > We would like to point out that human data might not be as scarce as the reviewer thinks. For example, **banks possess the personal data of the loan seekers** e.g., describing the same applicant applying again for a loan _after some time_. As evidence of this, please consider the following papers studying counterfactuals and fairness-related issues [1,2] using **confidential data from a large Italian bank** (Intesa San Paolo S.p.A.). Unfortunately, these data are proprietary and not easily accessible, which is a different problem and tangential to our contributions or to learning an estimator $P(\mathbf{X}^t)$.
> >
> > > **Time can invalidate recourses is also not something too insightful, in my opinion [...]**
> >
> > We believe the reviewer’s claim could be applied to many other studies considering the robustness of recourse (e.g., all the ones described in the _“Further related works”_ paragraph on page 3), but we reject the implication that they are not insightful. On the contrary, given that it is obvious that, without robustness, it would be _impossible_ to realize recourse practically, we believe it is an issue worth investigating extensively.
> >
> > We would like to argue that we took a step further from simple intuition by **grounding and studying** the problem through a **formal framework and empirical evaluations**. For us, it is insightful that current robust solutions to recourse do not work well with time (Section 3.4), that counterfactual recourse has even more pitfalls (Section 3.1), and that we can devise remedies to mitigate this issue (Section 3.6), thus prompting us to be more careful when considering realistic applications of recourse.
> >
> > > **Could the authors explain why varying lambda is meaningful? [...]**
> >
> > The value $\mathbf{x}^*$ (line 4) is the “worst-case” future user in the uncertainty set, e.g., the future user with the lowest score given by the classifier $h$. If $\lambda$ is high, then we will penalize the loss more if $\theta$ does not achieve recourse for $\mathbf{x}^*$ (line 5). As the reviewer pointed out correctly, this is _conservative_, since we are optimizing the $\theta$ achieving recourse for the worst-case scenario, thus increasing the overall cost, while our user might not turn out that “bad” in the future. Therefore, by decreasing $\lambda$, we can make our solution less conservative, since we do not penalize our loss so heavily for the worst-case future user. Hence, by varying $\lambda$ we can study in a principled way the tradeoff between validity (e.g., being very conservative or not) and cost as the reviewer asked in their initial review.
> >
> > While drafting the answer, we realized we missed a typo in Algorithm 1, where on line 4 it should be $\text{argmin}$, and not $\text{argmax}$. We will fix it in the revised version of the manuscript.
> >
> > [1] Crupi et al. "Counterfactual explanations as interventions in latent space." Data Mining and Knowledge Discovery (2024)
> >
> > [2] Castelnovo, Alessandro, et al. "Befair: Addressing fairness in the banking sector." IEEE BigData (2020)

---

> > > ### Author Response · Authors · 2024-11-25
> > > **Answer to Reviewer YDw8 (2/2)**
> > >
> > > > **I don't understand this argument. Why don't we wait until time $t+\tau'$ and recommend another recourse when we reach time $t + \tau'$? [...]**
> > >
> > > In order to answer as clearly as possible, we will start by reviewing how we formalise an action (and its duration) in our framework (Definition 1) through a fictional example.
> > >
> > > Let us imagine that, at time $t$, Alice is denied her loan application, and the bank tells her “You should get a degree”. Imagine it will take Alice $\tau$ timesteps to acquire the degree. Then, Alice will start studying for the degree at time $t$, and she will obtain it at time $t+\tau$.
> > >
> > > In reality, the _“get a degree”_ action is something that the user would **start** performing at time $t$ (e.g., enrolling at the university), which would **manifest** its causal effect at time $t+\tau$ (e.g., getting a degree).
> > >
> > > We formalize this behaviour by considering a simple causal model where we model the _” enrol at university/get a degree”_ action as a **single** intervention $\theta$ applied **directly** at time $t+\tau$, by assuming its causal effects manifest also at time $t+\tau$. Consequently, since any action done by users will take time to be implemented, we always consider any action to be performed at a later $t+\tau$, where $\tau$ is **always greater than zero** (see Definition 1).
> > >
> > > Now, to get back to the reviewer's comment, the alternative strategy of telling the user to _“wait until they are ready”_ is, unfortunately, infeasible since **any user action will be performed at a later time step than the one when the recourse is issued** (formalized as placing the constraint $\tau >0$).
> > >
> > > In our fictional example, even if Alice were ready to perform the updated recourse action (e.g., get a degree) at time $t+\tau$, it would be unrealistic for Alice to _immediately_ obtain a degree at time $t+\tau$. Practically, we argue that by the time Alice has implemented the action (at an instant $t+\tau’$ where $\tau’ > \tau$) it might not be enough for her to achieve recourse. Thus, when Alice comes back we might have to suggest a different action, e.g., _“increase her yearly income by 20000$”_, potentially leading to the same negative result.  In other words, this procedure is wasteful and - theoretically - might not converge.
> > >
> > > Following the discussion with the reviewer, we will revise the main manuscript by incorporating this example to extend the discussion on the “naive solution”, and to better describe how we formalize a user's action.

---

> > > > ### Author Response · Authors · 2024-12-01
> > > >
> > > > Dear Reviewer,
> > > >
> > > > As we approach the end of the discussion period, we want to follow up to ensure that our previous responses have fully addressed all your questions and concerns. Please do not hesitate to let us know if you have any further questions or unresolved issues that we can clarify. Thank you again!

---

### Author Response · Authors · 2024-11-18

We would like to thank all reviewers for their careful and insightful comments, which helped us improve our paper. Please, find a point-by-point response below and the **revised PDF** incorporating the reviewers' suggestions with additional experimental results and clarifications. The changes are marked in **red** within the manuscript.

---

### Meta-Review · Area_Chair_4qkp · 2024-12-23

**Metareview:**

I read the paper and all review/rebuttals.

The paper presents a way of setting up algorithmic recourse (AR) in the context of longitudinal causal models. This makes a lot of sense, as recourses take time to implement, and ultimately are meant to modify the state of the system in the future, even when coming from a counterfactual interpretation of changing a state that is already established. AR lacks longitudinal models (to the best of my knowledge) and I don't see much coming from the dynamic treatment regime (DTR) literature in causal inference that covers AR.

Having read the paper and being familiar with AR and DTRs, and causal inference in general, I still felt challenged to fully embrace the way the problem is framed. There is much in the control/causal Markov decision processes (MDPs) literature about the influence of dynamics on the timing and type of action we take to achieve a particular state of interest at a particular time, with more classical control problems being directly amenable to constrained optimization.

Say we take a purely control-based approach, where I choose an action $A^t$ *now* to be at a particular state in $X^{t + \tau}$ the future. (Here, I use $A$ instead of $\theta$ to denote actionable variables - I really appreciate that the paper distinguishes $\theta$ from $X$, but it's not unusual in DTRs to separate actionable from non-actionable variables in a very general way.) In DTRs, actionable variables can encode arbitrary soft interventions of $X$. If I'm an athlete, and classifier $\hat Y^t$ decided that with my running time $X^t$ I shouldn't be selected to be part of an Olympic training team, then I can enter a training regime now (and incur the corresponding costs), as indexed by $A^t$, so that in the next Olympics selection in $\tau$ time steps I'm expected to be ready with a good enough running time. Action $A^t$ can be a shift intervention in some current state variables (say, $X_1^t$ is my weekly leg workout time, and my new value of $A^t$ encodes that I should add 10 to my current $X_1^{t - 1}$ for the next $\tau' \leq \tau$ time steps). The policy $\pi_^t$ encodes whether $A^t$ was controlled as $do(A^t = a^t)$ or whether it was observational and uncontrolled - I can still infer the $do(\cdot)$ effect from observational data based on assumptions such as sequential ignorability etc.

All of the above is to say that it's a great direction to take AR toward dynamic causal models, but I found the framing hard to follow when there is already so much in the literature of sequential causal models from the point of view of pure reinforcement learning, control theory, or the causal community perspective of dynamic treatment regimes. I don't fully agree with several points of the reviewers, but I think that some of the points about framing are vindicated, despite fully acknowledging the positive points made by g3T6. It is very jarring to see a paper on longitudinal causal modeling without a single connection to the vast literature on dynamic treatment regimes that started in the 1980s (or even optimization of Markov Decision Processes in general) and is subject of popular textbooks such as Hernan and Robins's "Causal Inference: What If" - and I'm speaking as someone who is firmly in the machine learning community.

My conclusion: I thoroughly support the authors in their research program, but they will have a bigger impact, and be better understood not only within the machine learning community but the broader range of disciplines where AR is valued, if they add the extra effort of tapping as much as possible into the existing, and cross-disciplinary, immense literature on longitudinal causal modeling. I don't see in this literature almost anything explicit about AR. I think the authors will benefit much from trying to establish these links, as evidenced by most of the reviewers struggling to get into the finer points presented here (and we shouldn't write off their concerns, even if not perfectly formulated).

**Additional Comments On Reviewer Discussion:**

There was a constructive set of reviews and rebuttals debating the presentation and contribution. I side with many of the points presented by g3T6, but ultimately I don't think we should fragment away a contribution that taps into mainstream longitudinal causal modeling from the large literature which is already out there and which would be the effective on incorporating AR problems.

---

### Decision · Program_Chairs · 2025-01-22

Reject